# The role of OCO-3 XCO₂ retrievals in estimating global terrestrial net ecosystem exchanges

Xingyu Wang[1], Fei Jiang[1,2,5,*], Hengmao Wang[1], Zhengqi Zhang[1], Mousong Wu[1], Jun Wang[1], Wei He[4], Weimin Ju[1,5], Jing M. Chen[3,6]

[1]Jiangsu Provincial Key Laboratory of Geographic Information Science and Technology, International Institute for Earth System Science, Nanjing University, Nanjing, 210023, China.

[2]Jiangsu Center for Collaborative Innovation in Geographical Information Resource Development and Application, Nanjing, 210023, China.

[3]Department of Geography and Planning, University of Toronto, Toronto, Ontario M5S3G3, Canada.

[4]Zhejiang Carbon Neutral Innovation Institute, Zhejiang University of Technology, Hangzhou, Zhejiang 310014, China.

[5]Frontiers Science Center for Critical Earth Material Cycling, Nanjing University, Nanjing, 210023, China.

[6]School of Geographical Sciences, Fujian Normal University, Fuzhou, 350007, China

**\*Corresponding author: Fei Jiang (jiangf@nju.edu.cn)**

**Abstract**

Satellite-based column-averaged dry air $CO_2$ mole fraction ($XCO_2$) retrievals are frequently used to improve the estimates of terrestrial net carbon exchanges (NEE). The Orbiting Carbon Observatory 3 (OCO-3) satellite, launched in May 2019, was designed to address important questions about the distribution of carbon fluxes on Earth, but its role in estimating global terrestrial NEE remains unclear. Here, using the Global Carbon Assimilation System, version 2, we investigate the impact of OCO-3 $XCO_2$ on the estimation of global NEE by assimilating the OCO-3 $XCO_2$ retrievals alone and in combination with the OCO-2 $XCO_2$ retrievals. The results show that when only the OCO-3 $XCO_2$ is assimilated (Exp_OCO3), the estimated global land sink is significantly lower than that from the OCO-2 experiment (Exp_OCO2). The estimate from the joint assimilation of OCO-3 and OCO-2 (Exp_OCO3&2) is comparable on a global scale to that of Exp_OCO2. However, there are significant regional differences. Compared to the observed global annual $CO_2$ growth rate, Exp_OCO3 has the largest bias, and Exp_OCO3&2 shows the best performance. Furthermore, validation with independent $CO_2$ observations shows that the biases of the Exp_OCO3 are significantly larger than those of Exp_OCO2 and Exp_OCO3&2 at mid and high latitudes. The reasons for the poor performance of assimilating OCO-3 $XCO_2$ alone include the lack of observations beyond 52°S and 52°N, the large fluctuations in the data amount, and its varied observation time. Our study indicates that assimilating OCO-3 $XCO_2$ retrievals alone leads to an underestimation of land sinks at high latitudes, and that a joint assimilation of OCO-2 $XCO_2$ and the OCO-3 $XCO_2$ retrievals observed in the afternoon is required for a better estimation of global terrestrial NEE.

## 1 Introduction

The rising of the carbon dioxide ($CO_2$) concentration in the Earth's atmosphere in recent decades, which is mainly caused by human activities, such as the burning of fossil fuels, deforestation and land-use change, has become a global concern (Hansen et al., 2013). Terrestrial ecosystems and oceans together absorb about 56 % of anthropogenic $CO_2$ emissions (Friedlingstein et al., 2023). Among them, terrestrial ecosystems play a crucial role in regulating the atmospheric $CO_2$ concentration. However, the carbon uptake capacity of terrestrial ecosystems varies considerably globally and regionally (Bousquet et al., 2000; Takahashi et al.,2009; Piao et al., 2020). Therefore, accurate quantification of global and regional terrestrial net ecosystem exchange (NEE) is very important to understand their role and potential in regulating changes in the atmospheric $CO_2$ concentration.

Atmospheric inversion is a major method for estimating surface carbon fluxes from observations of atmospheric $CO_2$ concentration (Enting and Newsam, 1990; Gurney et al., 2002; Thompson et al., 2016; Jiang et al., 2021), but it is more effective at the global scale than at the regional scale. A large number of previous studies have shown that different atmospheric inversion models can produce relatively consistent global estimates of carbon fluxes, but their performance at regional scales is variable. In regions such as the tropics, southern hemisphere oceans, and most continental interiors (South America, Africa and boreal Asia), the reliability of atmospheric inversions varies considerably due to the heterogeneous distribution of *in-situ* observations, leading to an increase in the uncertainty of carbon flux estimates (Peylin et al., 2013; Wang et al., 2019). The use of satellite observations to constrain atmospheric inversions can be effective in improving carbon flux estimates because of their better spatial coverage (Basu et al., 2013; Byrne et al., 2020; Jiang et al., 2021; Wang et al., 2022; He et al., 2023a). The National Aeronautics and Space Administration (NASA) launched the Orbiting Carbon Observatory 2 (OCO-2) satellite in 2014 (Crisp et al., 2017; Eldering et al., 2012, 2017), followed by the Orbiting Carbon Observatory 3 (OCO-3) satellite in 2019 (Taylor et al., 2023). The OCO satellites have a high sensitivity to column-averaged dry air CO2 mole fraction ($XCO_2$), a fine footprint, and good spatial coverage, and can therefore be used to better constrain surface carbon flux estimates. In previous studies, many atmospheric inversion models have used the $XCO_2$ from the OCO-2 satellites to estimate global (e.g., Crowell et al., 2019; Peiro et al., 2022; Byrne et al., 2023) and regional (e.g.,

Palmer et al., 2019; Byrne et al., 2021; Philip et al., 2022; He et al., 2022; He et al., 2023a) surface
carbon fluxes. For example, Miller et al. (2018) evaluated the effectiveness of OCO-2 observations in
constraining regional biospheric $CO_2$ fluxes. Their findings indicate that OCO-2 observations are most
effective at continental and hemispheric scales. Byrne et al. (2022) utilised OCO-2 data to fill a gap in
station observations at high latitudes. Their study confirmed the presence of significant and widely
distributed early cold-season $CO_2$ emissions in the northeastern region of Eurasia. Furthermore, several
studies have utilised OCO-2 $XCO_2$ data to investigate the impact of climate extremes on terrestrial
NEE, such as El Niño (e.g., Liu et al., 2017) and droughts (He et al., 2023 b; Chen et al., 2024). OCO-
3 introduces new technologies and observational methods to monitor $CO_2$ on Earth, offering the same
spatial resolution as OCO-2. It is aimed at detecting mid-latitude regions where human $CO_2$ emissions
are concentrated. However, few studies have used the OCO-3 $XCO_2$ retrievals to constrain global and
regional surface carbon fluxes until now. Therefore, it is important to investigate the impact of assim-
ilating OCO-3 observations on the estimates of global and terrestrial carbon sinks.
In this study, we used both OCO-2 and OCO-3 $XCO_2$ retrievals to invert global and regional
carbon fluxes for the period of 2020-2022 with the Global Carbon Assimilation System, version 2
(GCASv2) (Jiang et al., 2021). The $XCO_2$ retrievals from OCO-2 and OCO-3 were assimilated sepa-
rately and together in order to disentangle the effect of OCO-3 $XCO_2$ retrievals on the estimates of
global and regional terrestrial carbon sinks.

**2 Methods and data**

**2.1 Inversion method**
The Global Carbon Assimilation System, version 2 (GCASv2) (Jiang et al., 2021; Wang et al.,
2021) designed primarily for assimilating satellite $XCO_2$ retrievals was adopted in this study to invert
surface carbon fluxes. The system uses the Model for Ozone and Related Chemical Tracers, version 4
(MOZART-4; Emmons et al., 2010) to simulate three-dimensional atmospheric $CO_2$ concentrations,
and an ensemble square root filter (EnSRF; Whitaker and Hamill, 2002) to implement the inversion of
surface fluxes. MOZART-4 is an offline global chemical transport model developed in the National
Center for Atmospheric Research (NCAR). It can be driven by essentially any meteorological data set
and with any emissions inventory, so there is not a unique standard simulation (Emmons et al., 2010).
We turned off all gas-phase, heterogeneous chemical reactions, aerosol and deposition processes in the
MOZART4 model and added a corresponding number of $CO_2$ tracers according to the ensemble num-
ber in GCASv2, in order to allow the model to run more quickly. EnSRF assimilates observations in a
sequential way, and obviates the need to perturb the observations. It shows good performance as long
as the observation errors are uncorrelated (Houtekamer and Mitchell, 2001). GCASv2 is an upgrade
from the GCAS (Zhang et al., 2015) that was established in 2015. The main upgrades include: 1) the
addition of an assimilation module for satellite observations; 2) a change in the assimilation algorithm
(i.e., EnSRF); 3) a change in the operational flow of the assimilation system; 4) the addition of a 'super-
observation' scheme; 5) inversion of fluxes at the grid scale; and 6) an improvement in the localization
scheme.

118        GCASv2 runs cyclically, with a two-step optimization strategy in each assimilation window (1

week). In the first step, the prior fluxes ($\boldsymbol{X_0^b}$) in each grid are independently perturbed with a random
number ($\delta_i$) drawn from a Gaussian distribution with mean of 0 and standard deviation of 1, and a
scaling factor ($\lambda$) that represents the uncertainty of each prior flux (Eq. 1).

$$\boldsymbol{X_i^b} = \boldsymbol{X_0^b} + \lambda \times \delta_i \times \boldsymbol{X_0^b} \ , \mathrm{i} = 1, 2, \dots , \mathrm{N} \tag{1}$$

Then, the perturbed fluxes are put into the MOZART-4 model to simulate ensembles of $CO_2$ concen-
trations. The $CO_2$ profiles are sampled according to the locations and times of $XCO_2$ observations and
converted to the simulated ensembles of $XCO_2$ ($XCO_{2,i}^m$) according to prior $XCO_2$ ($XCO_2^a$), prior $XCO_2$
profiles ($y_{a,j}$), pressure weighting function ($h_j$), and averaging kernel ($a_j$) of the $XCO_2$ retrievals (Eq.

127     2).

$$XCO_{2,i}^m = XCO_2^a + \sum_j h_j a_j (A(CO_{2,i}) - y_{a,j}) \tag{2}$$

Subsequently, the perturbed fluxes ($\boldsymbol{X_i^b}$), the simulated $XCO_2$ ensembles and the observed $XCO_2$ ($y$)
are used in EnSRF    to optimize the carbon fluxes ($\overline{\boldsymbol{X^a}}$) (Eqs. 3-5). The background error covariance
matrix ($\boldsymbol{P^b}$) is calculated based on $\boldsymbol{X_i^b}$ according to Eq. (3), where $\overline{\boldsymbol{X}}^b$ is the mean of $\boldsymbol{X_i^b}$. The pos-
terior flux ($\overline{\boldsymbol{X^a}}$) is a correction to the prior flux using the bias between simulated and observed $XCO_2$
($\mathbf{y} - \boldsymbol{H}\overline{\boldsymbol{X^b}}$) and the Kalman gain matrix ($\boldsymbol{K}$) (Eq. 4). And $\boldsymbol{K}$ is calculated according to Eq. (5), which is
a function of model-data mismatch error covariance matrix ($R$) and the background error covariance
matrix.

$$P^b = \frac{1}{n-1}\sum_{i=1}^{n}(X_i^b - \overline{X}^b)(X_i^b - \overline{X}^b)^T \tag{3}$$

$$\overline{X^a} = \overline{X^b} + K(y - H\overline{X^b}) \tag{4}$$

$$K = P^b H^T (H P^b H^T + R)^{-1} \tag{5}$$

In the second step, the optimized carbon fluxes are put into the MOZART-4 model to obtain the
initial field of the next assimilation window. This scheme allows compensation of inversion results
between neighboring windows and mass conservation between flux adjustments and concentration
changes.
In order to reduce the effects of horizontal observation error correlation and representativeness
error, based on the optimal estimation theory (Miyazaki et al., 2012), the system also performs a "su-
per-observation" scheme, which combines multiple observations located within a same model grid into
a single high-precision "super-observation". In this method, it first calculates the simulated $XCO_2$ cor-
responding to each observed $XCO_2$ based on the observation time and location, and then, it performs
a retrieval error-weighted average for all the simulated and observed $XCO_2$ falling within the same
model grid in the DA window, respectively.
There are inevitably spurious correlations in the EnKF method, to reduce the effect of spurious
correlations, a two-layer localization scale was adopted in GCASv2, which is used to select which
observations can be used for the flux analysis for each grid. The localization technique is based on the
correlation coefficient between the simulated $XCO_2$ ensembles ($XCO_{2,i}^m$) in each observation location
and the perturbed fluxes ($X_i^b$) in current model grids and their distances. The observations will be
accepted for assimilation if the distance is less than 500 km and the correlation coefficient is greater
than 0; and if the distance is greater than or equal to 500 km and less than 3000 km and the correlation
coefficient should be significant ($p<0.05$). Otherwise, the observations are not accepted. The reason
for this scheme is that considering the atmospheric horizontal diffusion, we believe that there must be
a correlation between the flux of one grid and the concentrations in its neighbouring grids, and there-
fore observations are accepted as long as this correlation coefficient is greater than zero. In contrast,
at distant locations (>500 km), where the effect of atmospheric horizontal diffusion is essentially neg-
ligible, the relationship between source and receptor is mainly due to atmospheric transport, and in
order to minimize spurious correlations we require that such correlations must be significant.More
details of the system can be found in Jiang et al (2021).

**2.2 OCO-2 and OCO-3 XCO$_2$ retrievals**

In July 2014, the Orbiting Carbon Observatory (OCO) -2 satellite was launched by NASA with
the primary objective of providing accurate space-based measurements to quantify changes in XCO$_2$.
The satellite is equipped with three high-resolution spectrometers that can detect two near-infrared
wavelength bands (1.61μm and 2.06 μm) of sunlight reflectance spectra to observe CO$_2$. In May 2019,
NASA launched OCO-3 to the International Space Station (ISS) to detect CO$_2$ in mid-latitudes, where
human emissions are more concentrated. OCO-3 operates in a low-inclination orbit from 52°S to 52°N
and is equipped with three high-resolution spectrometers, providing the same spatial resolutions and
similar observation mode as the OCO-2 satellite (Taylor et al., 2023). However, since OCO-3 is
mounted on the ISS, its observation time and frequency for the same place is different from the OCO-

175    2.

The XCO$_2$ data from OCO-3 and OCO-2 used in this study are bias-corrected products from
August 2019 to December 2022 at the image element level. The data are sourced from Version 10.4r
Level 2 Lite and Version 11.1r Level 2 Lite, respectively. Before using them in our inversion system,
it is essential to pre-process the data. First, both the land (Land Nadir + Land Glint, LNLG) and ocean
(Ocean Glint, OG) retrievals were adopted, and they were filtered using the parameter of XCO$_2$_qual-
ity_flag, which indicates the quality of the data. Only data with XCO$_2$_quality_flag=0 was selected for
assimilation in this study. Then, the LNLG and OG retrievals and their corresponding retrieval param-
eters (namely $XCO_2^a$, $y_{a,j}$, $h_j$, and $a_j$ in Eq. 2) were re-gridded to a spatial resolution of 1° × 1° and
5° × 5° using the arithmetic averaging method, respectively. For the OG data, we used a coarser re-
gridding resolution, that is because the distribution of XCO$_2$ is more homogeneous on sea than on land.
Finally, both OCO-3 and OCO-2 XCO$_2$ retrievals were converted to the X2019 scale of the World
Meteorological Organization (WMO) following Hall et al., (2021). Figure 1a and c display the distri-
bution and coverage of screened OCO-3 and OCO-2 $XCO_2$ retrievals from 2020 to 2022. Compared
to OCO-2, OCO-3 has more observational data in the mid-latitudes of the northern and southern hem-
ispheres, especially in arid and semi-arid regions.

191         Following Jiang et al. (2022), the model-data mismatch errors were amplified by a factor on top

of the $XCO_2$ posterior errors, but with the minimum observation error setting to 1 ppm. It needs to be
noted that in the OCO-3 and OCO-2 products, the $XCO_2$ posterior errors of OG retrievals ($0.48\pm0.11$
and 0.51±0.15 ppm in 2020 for OCO-2 and OCO-3, respectively) are smaller than LNLG ($0.54\pm0.12$
and 0.64±0.18 ppm in 2020 for OCO-2 and OCO-3, respectively), but in fact, the observational error
should be greater at sea than on land (Peiro et al., 2022). Therefore, before multiplying by a uniform
factor, we increased the $XCO_2$ posterior errors of OG retrievals by 0.2 ppm. Taylor et al. (2023) re-
ported that the mean of the uncertainties for the OCO-2 and OCO-3 quality-filtered and bias-corrected
$XCO_2$ are 1.0 and 1.3 ppm, respectively. Considering that the global atmospheric transport model may
have an uncertainty of about 1.0 ppm (Lauvaux et al., 2009), thus in this study, we set the amplification
factor to be 3.5. Through this treatment, the mean model-data mismatch errors of LNLG and OG are
about 1.9 and 2.4 ppm for OCO-2, and 2.3 and 2.5 ppm for OCO-3, respectively.

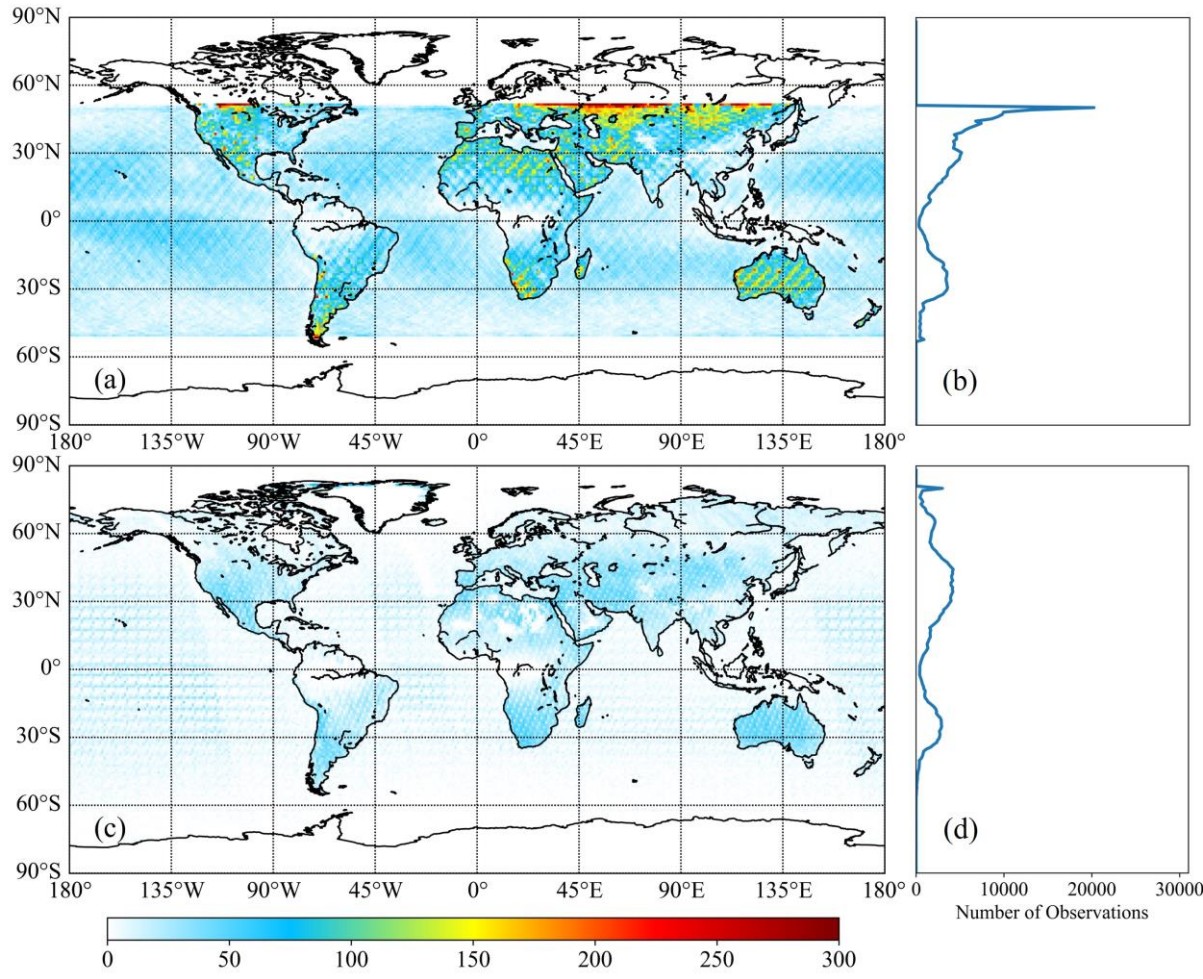


**Figure1.** Data amount (the sum of 2020-2022) of $XCO_2$ in each grid cell (1° × 1°) and at each latitude used
in this study (a, b, OCO-3; c, d, OCO-2)

## 2.3 Prior carbon fluxes

There are 4 prior carbon fluxes used in this study, which are terrestrial NEE, ocean-atmosphere
(OCN) carbon exchanges, fossil fuel and cement production (FOSSIL) carbon emissions, and biomass
combustion (FIRE) carbon emissions. The NEE were simulated using the BEPS model (Chen et al.,
2019). The OCN fluxes were derived from the mean of the JMA Ocean $CO_2$ Map (Iida et al., 2021),
which contains a global product with 1°×1° resolution (Globe, v2022) and another product for the
Northwest Pacific region with a resolution of 0.25°×0.25° (The western North Pacific, v2023). These
two products were integrated before they are used in this study. The FOSSIL carbon emissions were
obtained from GCP-GridFEDv2023.1 (Jones et al., 2021), which contains monthly global carbon emis-
sions from fossil fuels, cement production, and cement product weathering carbon sequestration at a
spatial resolution of 0.1º×0.1º. The FIRE carbon emissions were obtained directly from the Global
Fire Emissions Database, Version 4.1(GFED4.1s; Randerson et al., 2017). All 4 prior fluxes cover the
entire time period of this study (i.e., August 2019 to December 2022) and they were re-gridded to a
unified spatial resolution of 1º×1º before used in the GCASv2 system.
**2.4 Evaluation data and methods**
Due to the significant spatial scale discrepancy between the inverted fluxes and the *in-situ* ob-
served fluxes, direct validation of the posterior Net Ecosystem Exchange (NEE) using observed data
is typically unattainable. However, we are able to indirectly evaluate the posterior fluxes by comparing
the atmospheric $CO_2$ concentrations, simulated with the posterior fluxes, against independent $CO_2$
measurements. (e.g., Jin et al., 2018; Wang et al., 2019; Feng et al., 2020; Jiang et al., 2021). In this
study, we used surface flask observations at 66 sites from the ObsPack dataset (ObsPack v9.1, Schuldt
et al., 2023) to independently assess the posterior fluxes. The screening of the 66 sites followed the
methodology of Jiang et al. (2022). The distribution of the 66 flask sites is shown in Figure 2. The
specific metrics assessed were the statistics of mean bias (BIAS), absolute bias (MAE), and root mean
square error (RMSE). We calculated annual BIAS, MAE, and RMSE globally, for different latitudinal
zones, and for different land areas.

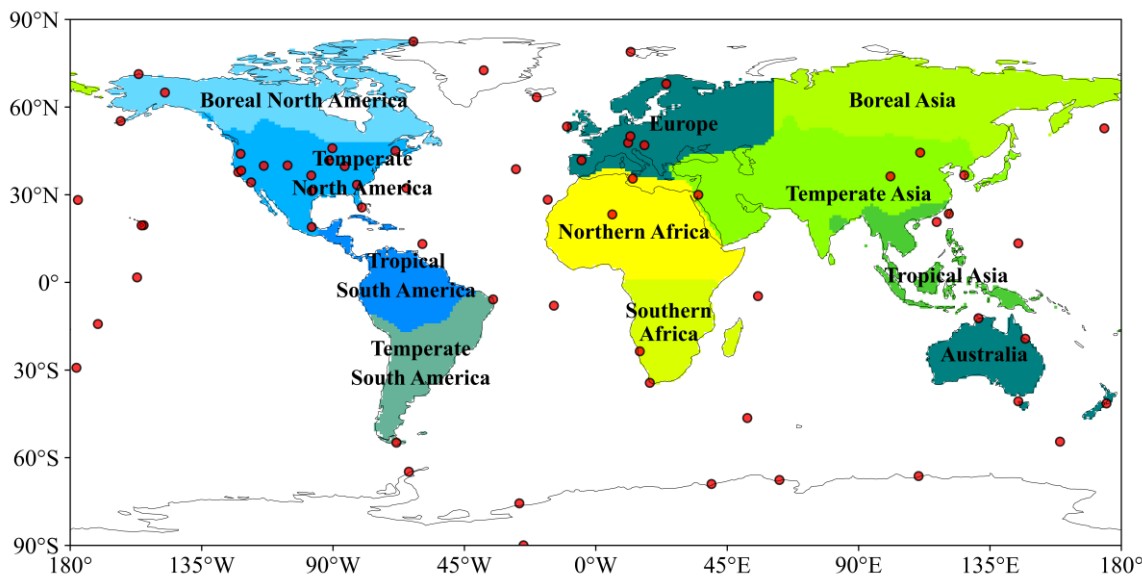


**Figure2.** Distributions of the observation sites used for independent evaluation in this study and the 11 Trans-
Com-3 regions on land defined in Botta et al. (2012).

## 3 Inversion experiments

The GCASv2 system was run from 1 August 2019 to 31 December 2022. The initial five months were designated as the spin-up stage, and the results from January 2020 to December 2022 were analyzed in this study. Three inversion experiments were conducted: (1) assimilation of OCO-3 $XCO_2$ (all inversion experiments use OG+LNLG data) retrievals alone (Exp_OCO3); (2) assimilation of OCO-2 $XCO_2$ retrievals alone (Exp_OCO2); and (3) simultaneous assimilation of OCO-3 and OCO-2 $XCO_2$ retrievals (Exp_OCO3&2). In each experiment, the methodology employed was consistent with that of previous studies (Peters et al., 2007; Jiang et al., 2021, 2022), only the NEE and OCN fluxes were optimized, and the FIRE and FOSSIL emissions are prescribed. According to Eq. (1), the prior NEE and OCN fluxes were perturbed using Eq. (6).

$$X_i^b = \lambda_{NEE} \times \delta_{i,NEE} \times X_{NEE}^b + \lambda_{ocn} \times \delta_{i,ocn} \times X_{OCN}^b + X_{Fire}^b + X_{Fossil}^b, i = 1, 2, ..., N \quad (6)$$

where $X_{NEE}^b$, $X_{OCN}^b$, $X_{Fire}^b$, and $X_{Fossil}^b$ represent the prior fluxes of NEE, OCN, FIRE, and FOSSIL, respectively; $\delta_i$ is random perturbation samples, which is independent between grids; $\lambda_{NEE}$ and $\lambda_{ocn}$ are the scaling factors for prior NEE and OCN fluxes, which were set to be 6 and 10 in this study, respectively. As described above, the prior fluxes have a spatial resolution of $1° \times 1°$, for $\delta_{i,NEE}$ and $\delta_{i,ocn}$, we adopted a spatial resolution of $3° \times 3°$, and the outputs of the posterior fluxes have the same spatial resolution with the prior fluxes, that means in each $3° \times 3°$ grid, the prior fluxes were adjusted with a same factor.

Additionally, two forward simulations were conducted to obtain the prior and posterior $CO_2$ concentrations, which were then compared with the independent $CO_2$ observations to assess the posterior carbon fluxes. Following Jiang et al. (2022), MOZART-4 is driven by the $1.9° \times 2.5°$ grids version of the GEOS5 Global Atmosphere Forcing Data (Tilmes, 2016). It has a vertical level of 72 layers, and MOZART-4 uses the lowest 56 vertical levels of GEOS-5 and the same spatial resolution with GEOS-5 data.

## 4 Results and discussion

### 4.1 Global carbon budget

Table 1 presents the prior and the posterior annual global carbon budgets from the 3 inversion

experiments during 2020-2022. The global terrestrial NEEs obtained from the Exp_OCO3, Exp_OCO2, and Exp_OCO3&2 experiments are -3.41±0.65, -4.17±0.60, and -4.14±0.57 PgC yr$^{-1}$, respectively. The global NEE inferred from the Exp_OCO3 is significantly weaker than those from Exp_OCO2 and Exp_OCO3&2, and the latter two are comparable. For the OCN carbon sink, Exp_OCO3 has the strongest sink but is closest to the a priori result, while Exp_OCO2 and Exp_OCO3&2 have essentially the same sink. Combined with the FOSSIL and FIRE carbon emissions, the global net carbon fluxes are 4.74±0.77, 5.55±0.67, 4.90±0.63, and 4.93±0.60 PgC yr$^{-1}$ for the a priori, Exp_OCO3, Exp_OCO2, and Exp_OCO3&2, respectively. In comparison with the average atmospheric $CO_2$ growth rate of 4.93 PgC yr$^{-1}$ for 2020-2022 given by the Global Carbon Budget 2023 (Friedlingstein et al., 2023), the results of Exp_OCO3&2 are the closest, with a mean bias of 0.0 PgC yr$^{-1}$, whereas Exp_OCO3 has the largest bias, with a deviation of 0.62 PgC yr$^{-1}$. This indicates that the carbon sinks in Exp_OCO3 may be significantly underestimated, and joint assimilation of OCO-2 and OCO-3 $XCO_2$ retrievals gives the best performance on a global scale.

**Table 1.** Global carbon budget estimated in the 3 inversion experiments (PgC yr$^{-1}$).

|  | Prior | Exp_OCO3 | Exp_OCO2 | Exp_OCO3&2 |
|---|---|---|---|---|
| FOSSIL emissions | | 9.71 | | |
| FIRE emissions | | 1.97 | | |
| NEE | -4.10±0.75 | -3.41±0.65 | -4.17±0.60 | -4.14±0.57 |
| OCN fluxes | -2.84±0.17 | -2.71±0.17 | -2.61±0.17 | -2.61±0.17 |
| Global net carbon fluxes | 4.74±0.77 | 5.55±0.67 | 4.90±0.63 | 4.93±0.60 |
| Observed global $CO_2$ growth rates | | 4.93 | | |

## 4.2 Regional NEE

Figure 3 shows the spatial distribution of annual mean posterior terrestrial fluxes and oceanic fluxes from the Exp_OCO3, Exp_OCO2, Exp_OCO3&2 and their differences against the a priori fluxes. Overall, the spatial distribution of carbon sources and sinks in terrestrial ecosystems obtained from different experiments is basically the same, with sources in western North America (N. America),

eastern Amazonia, parts of Siberia, parts of Northwest China, central and western Australia, and the
Sahel region and eastern parts of Africa, while other areas are carbon sinks. However, the carbon
sources/sinks obtained from Exp_OCO3 exhibit a markedly different strength compared to those de-
rived from the other two experiments. Compared with the prior flux, the terrestrial carbon sinks in
northeastern China, most of Europe, northern Siberia, the central and northeastern United States (US),
and southern Africa increased significantly in all the 3 experiments. However, the increase in terrestrial
carbon sinks in regions other than northeastern China in the Exp_OCO2 and Exp_OCO3&2 was
greater than that in the Exp_OCO3. Meanwhile, in southern Canada, western and southern US, eastern
Brazil and northern South America (S. America), the Sahel region and eastern parts of Africa, all the
3 inversion experiments show a significant decrease in the terrestrial carbon sink. The degree of change
in the inversion results is more pronounced in the Exp_OCO2 and Exp_OCO3&2 than in the
Exp_OCO3. Figure 3 also show the distribution of terrestrial carbon fluxes along latitudes. The poste-
rior and prior fluxes have a similar distribution trend along the latitude, with a significant peak of
carbon sink near 60°N, and the strongest sinks of Exp_OCO2 and Exp_OCO3&2 are comparable,
which are significantly stronger than the a priori, while Exp_OCO3 has the weakest peak of carbon
sink and that is close to the a priori. In addition, it also could be found that the terrestrial carbon sinks
obtained from Exp_OCO3 are also significantly smaller than those from Exp_OCO2 and
Exp_OCO3&2 near 30°S.

300         In order to better understand and compare the differences among different inversion experiments,

we have aggregated the prior and the posterior NEEs into the 11 TransCom-3 land regions (Figure 2),
as shown in Table 2. It is clearly that almost all terrestrial regions behave as carbon sinks, both prior
and posterior fluxes. Among the experiments, only the terrestrial NEE in northern Africa obtained by
Exp_OCO3&2 shows a weak carbon source. There is relatively good agreement between all the inver-
sion experiments on whether the land carbon flux is a source or sink, but there is significant difference
in the NEE values. In all regions except temperate N. America, northern Africa, temperate Asia, and
Australia, Exp_OCO3 shows a weaker carbon sink than Exp_OCO2. Comparing Exp_OCO3 with
Exp_OCO3&2, Exp_OCO3&2 shows stronger carbon sinks in temperate N. America, southern Africa,
Australia, and Europe; and weaker sinks in tropical S. America, northern Africa, and boreal Asia; and
elsewhere Exp_OCO3&2 shows sinks intermediate to the other two experiments.

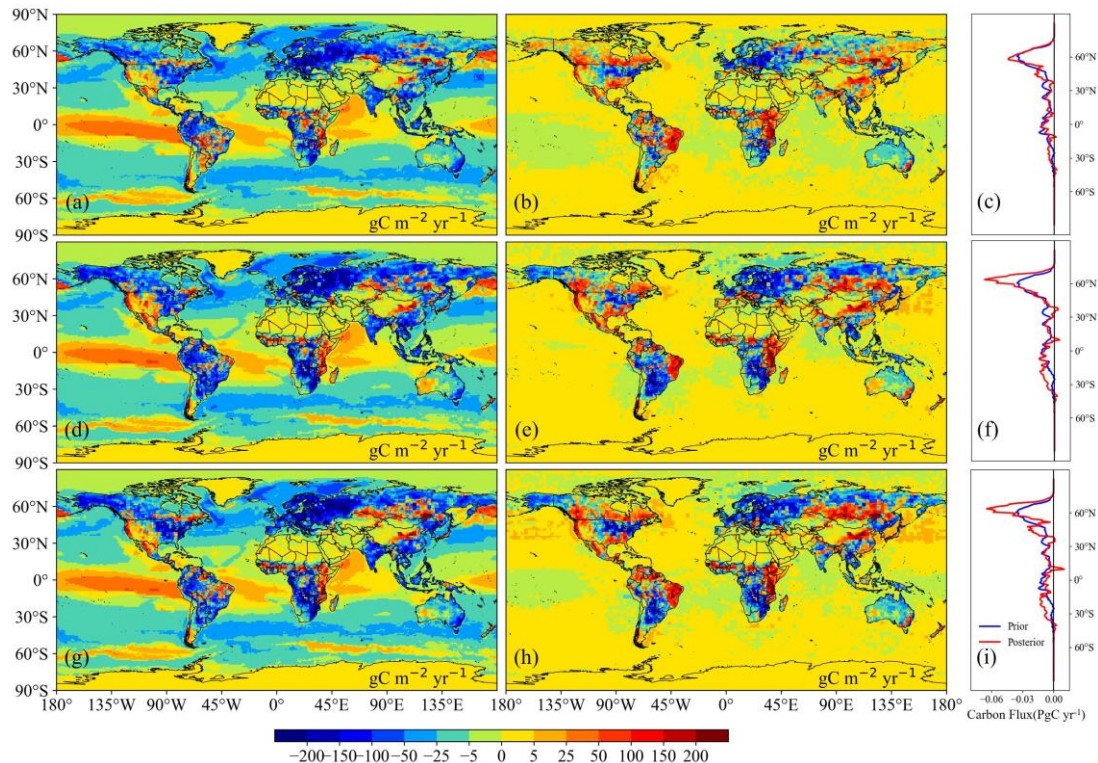


**Figure 3.** Spatial distribution of annual mean posterior terrestrial and oceanic carbon fluxes from 2020 to 2022,
the difference between posterior and prior fluxes, and the distribution of terrestrial NEEs at different latitudes.
(a, b, c, Exp_OCO3; d, e, f, Exp_OCO2; g, h, i, Exp_OCO3&2)


The regions with more pronounced differences among experiments are temperate S. America and
Europe. In Europe, the posterior fluxes of each inversion experiment show a pronounced carbon sink,
which is significantly larger than the prior flux, but the results of different experiments vary to some
extent , with NEEs ranging from -0.88±0.24 to -1.18±0.21 PgC yr$^{-1}$ (Table 2), with Exp_OCO3&2
having the largest sink. In the temperate S. America, Exp_OCO3 exhibits a very weak carbon sink,
whereas both Exp_OCO2 and Exp_OCO3&2 show a moderate carbon sink. One potential explanation
for this discrepancy is that the XCO$_2$ concentration observed by OCO-3 in the temperate South Amer-
ica is higher than that observed by OCO-2 for the duration of the study period (by ~0.55 ppm). Con-
sequently, in that assimilating the OCO-3 observations yields a weaker carbon sink. Compared with
the prior flux, the posterior NEE in the tropical S. America shows a significant discrepancy, the prior
flux show a very strong carbon sink of -0.78±0.23 PgC yr$^{-1}$, whereas the subsequent application of

constraints from satellite observations resulted in a reduction of the carbon sinks by approximately 2 to 3 times, with values ranging from -0.21±0.19 to -0.41±0.20 PgC yr$^{-1}$.

Following the imposition of constraints derived from satellite observations, the carbon sinks on the Northern Hemisphere land are all enhanced, with the largest enhancement of 0.59 PgC yr$^{-1}$ in Exp_OCO3&2, followed by 0.19 and 0.36 PgC yr$^{-1}$ in Exp_OCO3 and Exp_OCO2, respectively. While in the tropics, the carbon sinks were all weakened, with Exp_OCO3 being weakened most, by 0.67 PgC yr$^{-1}$, and the Exp_OCO2 and Exp_OCO3&2 being weakened by 0.37 and 0.59 PgC yr$^{-1}$, respectively; on Southern Hemisphere land, in Exp_OCO3, the sinks were weakened by 0.2 PgC yr$^{-1}$, whereas in Exp_OCO2 and Exp_OCO3&2, they were enhanced by 0.08 and 0.05 PgC yr$^{-1}$, respectively.

**Table 2.** Annual mean terrestrial fluxes (PgC yr$^{-1}$) in 2020-2022 for 11 TransCom-3 land regions, as well as for Northern Hemisphere land, Tropical land and Southern Hemisphere land. Includes the prior flux and the posterior fluxes from three inversion experiments.

| Regions | Prior | Exp_OCO3 | Exp_OCO2 | Exp_OCO3&2 |
|---|---|---|---|---|
| Boreal North America | -0.32±0.16 | -0.26±0.14 | -0.38±0.13 | -0.32±0.13 |
| Temperate North America | -0.19±0.30 | -0.25±0.25 | -0.12±0.25 | -0.35±0.21 |
| Tropical South America | -0.78±0.23 | -0.31±0.21 | -0.41±0.20 | -0.21±0.19 |
| Temperate South America | -0.28±0.22 | -0.03±0.17 | -0.40±0.16 | -0.27±0.14 |
| Northern Africa | -0.17±0.28 | -0.06±0.24 | -0.02±0.23 | 0.03±0.20 |
| Southern Africa | -0.30±0.24 | -0.30±0.19 | -0.49±0.17 | -0.54±0.16 |
| Boreal Asia | -0.56±0.26 | -0.37±0.24 | -0.52±0.21 | -0.34±0.23 |
| Temperate Asia | -0.42±0.23 | -0.33±0.20 | -0.22±0.19 | -0.30±0.18 |
| Tropical Asia | -0.37±0.13 | -0.31±0.12 | -0.39±0.11 | -0.35±0.11 |
| Australia | -0.15±0.09 | -0.20±0.08 | -0.11±0.08 | -0.21±0.07 |
| Europe | -0.40±0.26 | -0.88±0.24 | -1.01±0.19 | -1.18±0.21 |
| Northern Hemisphere lands | -1.89±0.56 | -2.08±0.49 | -2.25±0.44 | -2.48±0.44 |
| Tropical lands | -1.65±0.45 | -0.98±0.38 | -1.28±0.37 | -1.06±0.34 |
| Southern Hemisphere lands | -0.43±0.24 | -0.23±0.18 | -0.51±0.17 | -0.48±0.15 |

**4.3 Seasonal cycle of NEE**

Figure 4 illustrates the seasonal cycle of NEE for each TransCom-3 region. The posterior NEEs of different experiments are in good agreement on the seasonal cycle in most regions. In the Northern Hemisphere, the seasonal cycles of NEE in boreal N. America, temperate N. America, boreal Asia, temperate Asia, and Europe show relatively consistent trends. Carbon sinks in these regions generally occur from May to September and carbon sources from October to April. Large differences are evident in the strength of the carbon sinks observed in different regions, with different months in which the strongest carbon sinks occur. Boreal N. America, temperate N. America, and boreal Asia have the strongest carbon sinks in July, temperate Asia has the peak in July or August, and Europe has the strongest sinks in June. In the Southern Hemisphere, the southern Africa and temperate S. America have more consistent seasonal cycles, with their carbon sources occurring roughly from July to December and sinks from January to June. The strongest carbon sources all occur in October, and the strongest sinks occur around March. In Australia, carbon sinks occur mainly from March to October, with the peak occurring in August. In the tropics, southern Africa shows a seasonal cycle opposite to that of northern Africa, and carbon sinks occur from January to July with the strongest carbon sinks occurring near March. Tropical Asia shows a carbon sink in most months, with the strongest sink in September. The seasonal cycle in tropical S. America is more complex, with the strongest carbon source in October. In general, seasonal amplitudes are small in the tropics and large in the northern regions. The averaged seasonal amplitudes of the three inversion experiments in the boreal Asia, Europe, and temperate N. America are 1.17, 0.97, and 0.72 PgC yr$^{-1}$, respectively, while the seasonal amplitudes in tropical Asia and S. America are about 0.10 PgC yr$^{-1}$.

The regions where the difference between the prior and posterior NEEs is particularly pronounced are tropical S. America, southern Africa, Australia, and Europe. In the tropical S. America, the prior NEE is a significant sink from May to July, but after constraints from satellite observations, the carbon sink decreases significantly, even approaching neutral in June and July, and furthermore, in September and October, the sink also decreases significantly compared to the a priori. In southern Africa, the carbon sink is significantly stronger from January to March compared to the a priori, and conversely, the carbon source is significantly stronger in October and November. In Australia, the carbon sink is

significantly increased from January to August and decreased in October and November compared to the a priori. In Europe, there is a significant increase in the carbon sinks from May to June compared to the a priori.

As described in Section 4.2 that in temperate N. America, northern Africa, temperate Asia, and Australia, Exp_OCO3 shows a stronger sink than Exp_OCO2, which mainly occurs in May and June in temperate N. America, in August and September in northern Africa, from April to September in temperate Asia, and in Australia except for July. In other regions, Exp_OCO3 has weaker sinks than Exp_OCO2. In the high latitudinal regions, on the one hand, the carbon sinks in June and July of the Exp_OCO3 are generally smaller than those of Exp_OCO2, and on the other hand, the carbon source in October is significantly higher than that of Exp_OCO2, while in the tropics, the carbon sink is lower than that of Exp_OCO2 almost all year round. Compared to Exp_OCO3, Exp_OCO3&2 shows stronger carbon sinks in temperate N. America, southern Africa, Australia, and Europe, mainly in summer; and weaker sinks in tropical S. America, northern Africa, and boreal Asia,mainly in autumn. Elsewhere Exp_OCO3&2 shows carbon sinks intermediate to the other two experiments.

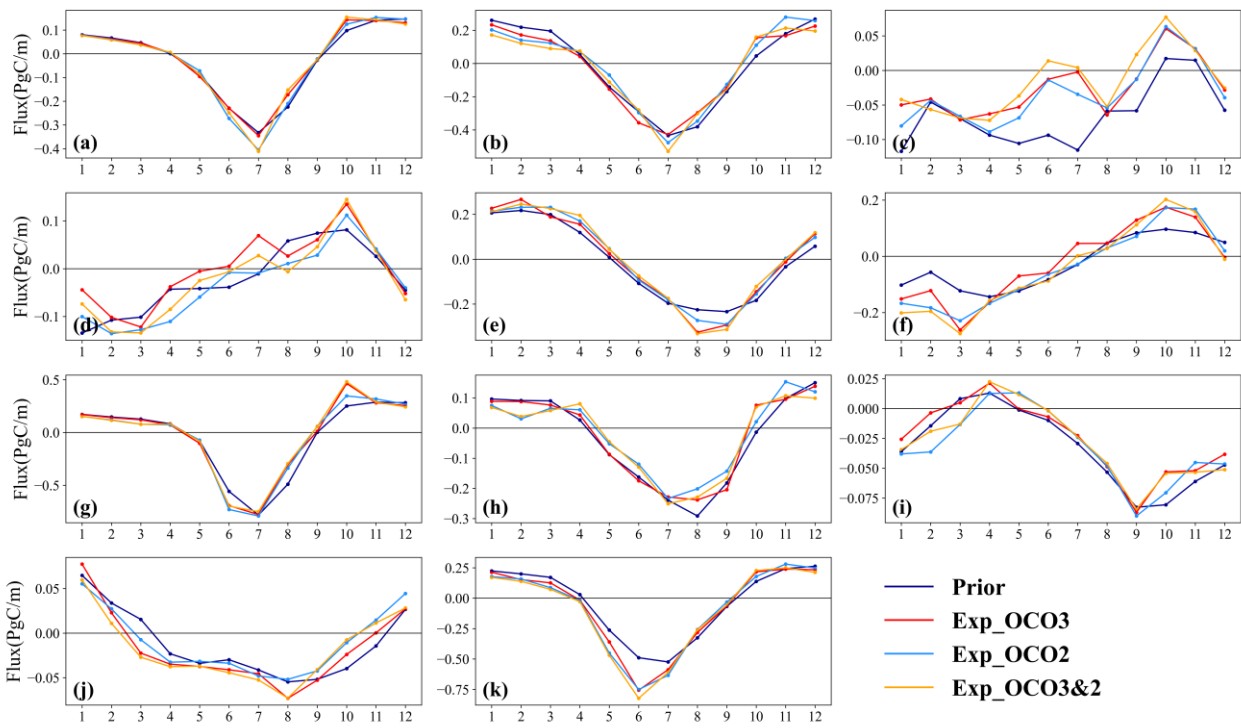

**Figure 4.** Averaged prior and posterior seasonal cycle of NEE in different TransCom-3 regions during 2020– 2022;(a) boreal N. America, (b) temperate N. America, (c) tropical S. America, (d) temperate S. America, (e)

northern Africa, (f) southern Africa, (g) boreal Asia, (h) temperate Asia, (i) tropical Asia, (j) Australia, (k) Europe.

## 4.4 Evaluation against independent observations

As shown in Figure 5, observations from 66 surface flask sites were used to evaluate the posterior fluxes. The prior and posterior $CO_2$ concentrations were simulated by the MOZART-4 model using the corresponding prior and posterior fluxes, as described in Section 3. The overall assessment results of the individual inversion experiments on a global scale are shown in Table 3. The results show that the mean BIAS, MAE, and RMSE between the prior $CO_2$ concentrations and surface flask observations are -1.82, 3.27, and 5.01 ppm, respectively. The prior BIAS shows a pronounced negative bias, which can be attributed to the fact that the prior NEE in 2019 (generated by the spin-up stage) was, on average, approximately 3.5 PgC less than the posterior NEE. This part of the NEE has an impact on the subsequent inversion. After constraints using the $XCO_2$ retrievals, the biases of the three experiments are reduced significantly compared to the a priori, indicating that the surface carbon fluxes have been improved. A comparison of the three inversion experiments reveals that Exp_OCO3 exhibits the largest BIAS, while Exp_OCO3&2 exhibits the lowest MAE and RMSE.

**Table 3.** Error statistics between the simulated $CO_2$ concentrations and surface flask observations (ppm).

|  | BIAS | MAE | RMSE |
| --- | --- | --- | --- |
| Prior | -1.82 | 3.27 | 5.01 |
| Exp_OCO3 | 0.32 | 2.44 | 4.56 |
| Exp_OCO2 | 0.02 | 2.42 | 4.49 |
| Exp_OCO3&2 | 0.05 | 2.34 | 4.47 |

Figure 5a and 5b illustrate the BIAS of the individual inversion experiments at different latitudinal zones and in different TransCom-3 land regions. In all latitudinal bands and all land regions, the $CO_2$ concentrations modelled by the a priori fluxes have the largest negative BIAS, which is greater than -1.2 ppm in all cases. Across latitudinal zones, in the Southern Hemisphere, and south of 30°N latitude, the Exp_OCO3 had the smallest BIAS, which is smaller than the Exp_OCO2 and comparable to the

results of the Exp_OCO3&2. However, in the mid to high latitudes of the Northern Hemisphere, the
BIAS of the Exp_OCO3 is higher than those of the Exp_OCO2 and Exp_OCO3&2. Especially in the
region north of 60°N latitude, the Exp_OCO3 exhibits a significant positive BIAS, while the
Exp_OCO2 and Exp_OCO3&2 both exhibit small negative BIAS. This suggests that the carbon sinks
at mid to high latitudes were underestimated. We also find that the OCO-3 retrievals help with the lack
of space-based $XCO_2$ observations in the tropics compared to OCO-2. The BIAS of Exp_OCO3&2 is
smaller than Exp_OCO2 in the region from 30°S to 30°N. Meanwhile, the BIAS of Exp_OCO3&2 is
also smaller than Exp_OCO2 in southern Africa, northern Africa and tropical Asia. Furthermore, we
can find that the BIAS can be further reduced in the mid to high latitudes of the Northern Hemisphere
after the addition of assimilated OCO-3 observations compared to the Exp_OCO2. In different Trans-
Com-3 land regions, the BIAS of the three inversion experiments is less than ±0.6 ppm, except in the
temperate Asia. In Africa, temperate S. America, tropical Asia, and Australia, the Exp_OCO3 had the
smallest BIAS, while the BIAS of Exp_OCO3&2 was between those of Exp_OCO3 and Exp_OCO2.
However, in temperate N. America and Europe, the Exp_OCO3 has the largest BIAS, followed by the
Exp_OCO2, while the Exp_OCO3&2 has the smallest BIAS.

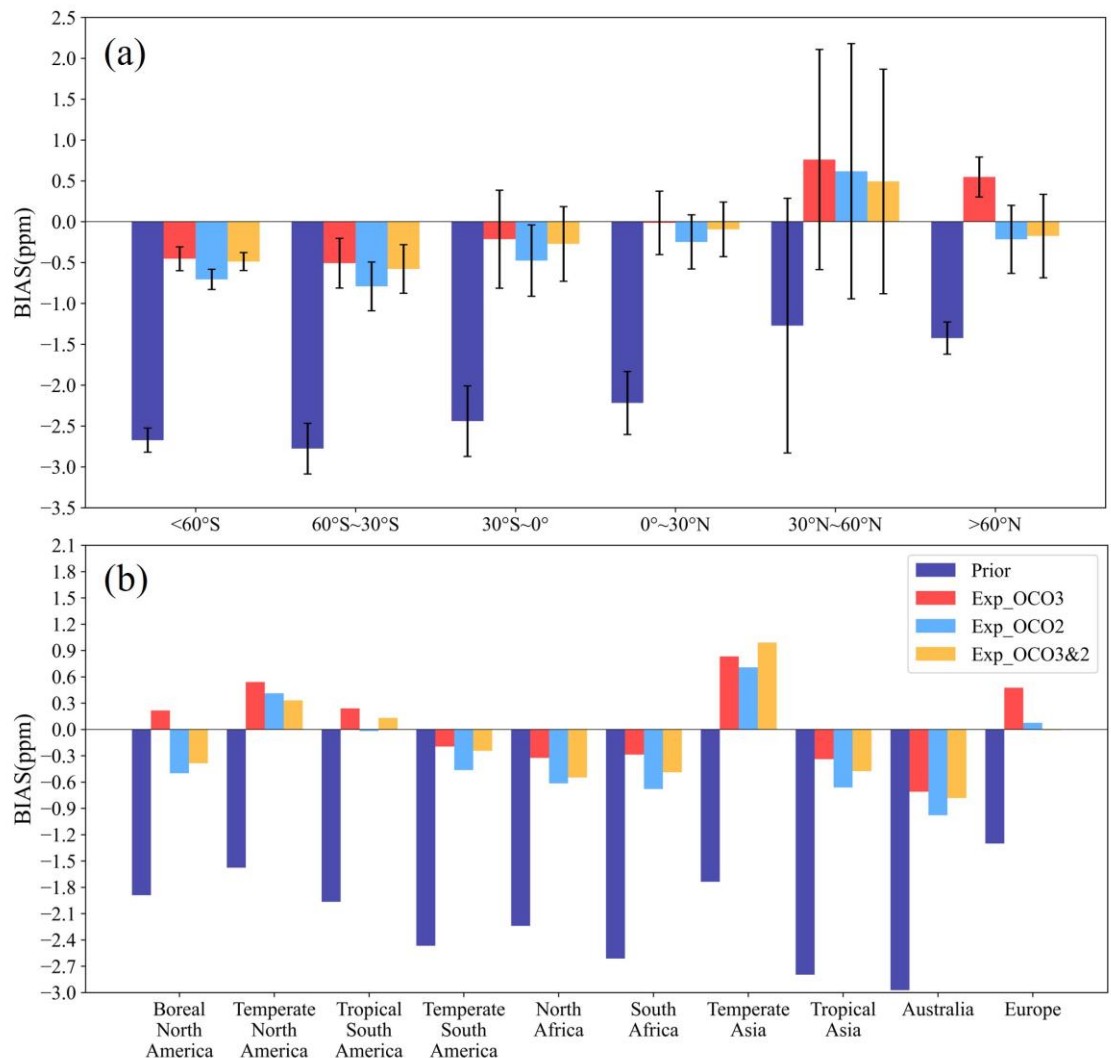

**Figure 5**. The prior and posterior $CO_2$ BIAS(a) at different latitudinal zones and (b) in different land regions.

**4.5 Discussion**

In most of the previous studies that used OCO-2 $XCO_2$ to invert surface carbon fluxes, the OG data were not used (e.g., Peiro et al., 2022; Byrne et al., 2023), the reason is that the OG $XCO_2$ may have larger uncertainties, inversions assimilating OCO-2 OG retrievals produced unrealistic results of annual global ocean sinks (Peiro et al., 2022). In addition to its large uncertainties, we believe that another reason for the poor assimilation performance of OG is the relatively homogeneous distribution of $XCO_2$ on ocean, causing a large correlation of the model-data biases among different $XCO_2$ observations within a same region, which leads to observations at the same region having the same direction

of adjustment for surface fluxes, and thus leads to a significant overestimated or underestimated ofocean carbon sink. Because of this, some assimilation algorithms (e.g., EnSRF) can only achieve better assimilation results when the model-data biases between observations have relatively small correlation or are uncorrelated. Therefore, in this study, we set the OG data with larger uncertainties than the LNLG data, and re-grided it at a coarser spatial resolution of $5° \times 5°$. The results show that under this scheme, the inverted ocean sink is reasonable, with value of -2.6 PgC yr$^{-1}$ (Table 1). In addition, in order to compare the scheme that we have adopted in this study with the previous scheme that do not assimilate the OG, we added three additional inversion experiments, in which only the LNLG data were assimilated (Table S1). It could be found that all the three inversion experiments without OG observations place smaller constraints on the ocean fluxes compared to the original experiments, with the posterior ocean fluxes remaining almost identical to the prior ocean fluxes. Correspondingly, the inverted global land sink as well as the sinks in most regions show a slight decrease (Tables S2 and S3). Evaluations in comparison with *in-situ* observations showed that there are some increases in the a posteriori concentration biases for all three experiments after removing OG. For example, for the experiments assimilating OCO-2 data, the mean bias increased from 0.02 to 0.14 ppm (Table S4). This suggests that assimilating OG data with our method can improve the inversions somewhat compared to removing OG.

Since OCO-3 has similar observation uncertainties of $XCO_2$ with OCO-2 (Taylor et al., 2023), the poor performance of assimilating OCO-3 $XCO_2$ retrievals (Exp_OCO3) may be related to that 1) OCO-3 lacks observations beyond 52° North and South latitudes (Figure 1a); 2) the observation time different from OCO-2; and 3) its spatial coverage between 52°S and 52°N. We first examined weekly changes in the data amount of OCO-3 using the re-grided data as described in Section 2.3, and found that there are very significant cyclical fluctuations in the data amount from OCO-3 (Figure S4a). Every 8 weeks or so, there is a trough in the data amount. There is a difference of about 5 times between the weeks with the highest and the lowest data amount, and in the weeks with least data amount, there were essentially no observations in the northern hemisphere (Figure S4b). This implies that the surface

carbon fluxes are largely unconstrained in the Northern Hemisphere, especially at mid- to high-latitudes, during the weeks with low observational data, resulting in poorer assimilation performance than for OCO-2. For the observation time, all observations of OCO-2 were at 1:30 p.m. local time (LST), whereas that of OCO-3 were variable, with only about 14% of the observations near 13:30 p.m. LST and about 54% in the morning or after 4:00 p.m. LST (Figure S1). For reasons such as coarser model resolution, the global atmospheric chemical transport models generally simulate atmospheric concentrations better only in the afternoon, when boundary layer heights are at their highest and atmospheric mixing is at its best, so assimilating these observations in the morning and after 4 p.m. LST may result in poorer inversions due to the greater simulation bias of the atmospheric transport models at these times of day.

In order to quantify these effects, we added another 3 additional inversion experiments, which were named as Exp_OCO2r, Exp_OCO3tc, and Exp_OCO2ts (Table S1). In Exp_OCO2r, only the OCO-2 $XCO_2$ retrievals located between 52°S and 52°N retrievals were assimilated, in Exp_OCO3tc, all the observation times of the OCO-3 $XCO_2$ retrievals were changed to 1.30 p.m. LST, and in Exp_OCO3ts, only OCO-3 data with observation times between 12 and 3 p.m. LST were assimilated. When the OCO-2 data beyond 52° North and South latitudes were also removed (Exp_OCO2r), the NEE estimates, both globally and for individual regions, are close to those of the Exp_OCO3 experiment, especially in the high latitude region of Europe and boreal North America, the inverted NEEs are almost identical to those of the Exp_OCO3 experiment (Table S2 and S3), and the bias of a posteriori concentrations from observations at high latitudes is close to that of the OCO-3 experiment (Figure S3). However, globally, compared to the OCO-3 experiment, the Exp_OCO2r experiment still has smaller the deviation between the global net flux and the observed annual growth rate (Table S2), and smaller the global mean bias of the posterior concentrations (Table S4). This suggests that the lack of observations of OCO-3 beyond 52° North and South latitudes does have a significant impact on the inversion results. In addition, it can also be noted that at mid-latitudes, the bias of Exp_OCO2r is also smaller than the OCO-3 experiment, which may be caused by the significant fluctuations in the data

amount of OCO-3 (Figure S4). When we changed all the observation times of the OCO-3 $XCO_2$ re-
trievals to 1.30 p.m. LST (Exp_OCO3tc), although we are not actually able to do so, the inversion does
show a significant improvement compared to Exp_OCO3. However, if we only select the data with
observation time between 12:00 and 3:00 p.m. LST (Exp_OCO3ts), the deviation between the global
net flux and the observed annual growth rate, and the mean biases of the posterior concentrations at
most latitudes are larger than those of Exp_OCO3 (Table S2 and Figure S3), indicating a poorer per-
formance than Exp_OCO3. The probably reason is that the data number of observations is substantially
reduced at this time (Figure S2), which leads to a substantial weakening of the observational constraints
on surface carbon fluxes (Figure S5).

**5 Summary and Conclusion**

501         In this study, we constrained terrestrial NEEs for the period from 1 August 2019 to 31 December

2022 using the OCO-2 and OCO-3 $XCO_2$ retrievals and the GCASv2 system, and analyzed the inver-
sion results from 2020 to 2022. We conducted three inversion experiments for separately and jointly
assimilating the OCO-2 and OCO-3 $XCO_2$ retrievals, to explore the impact of the OCO-3 $XCO_2$ re-
trievals on the constraints of global terrestrial NEEs. The prior and posterior $CO_2$ mixing ratios ob-
tained from forward simulations using the prior and posterior fluxes are analysed in comparison with
observations from 66 surface flask sites.

508         Globally, the terrestrial carbon sink from the Exp_OCO3 is smaller than the prior, while the ter-

restrial carbon sinks from the other two inversion experiments are slightly larger than the prior, but the
difference is small. The global net carbon flux from the Exp_OCO3&2 is very close to the observed
atmospheric $CO_2$ growth rate. Regionally, the posterior NEEs for most terrestrial regions show a car-
bon sink, with Europe showing a very strong sink and North Africa close to carbon neutrality. In the
Northern Hemisphere, the carbon sinks are enhanced, with the Exp_OCO3&2 being the most enhanced
by 0.59 PgC $yr^{-1}$ and the Exp_OCO3 and Exp_OCO2 by 0.19 and 0.36 PgC $yr^{-1}$, respectively. In the
tropics, the carbon sinks are weakened, with the Exp_OCO3 being the most weakened by 0.67 PgC
$yr^{-1}$, and the Exp_OCO2 and Exp_OCO3&2 sinks being weakened by 0.37 and 0.59 PgC $yr^{-1}$, respec-
tively; in the southern land, the sink inverted in Exp_OCO3 is weakened by 0.2 PgC $yr^{-1}$, whereas
those in the Exp_OCO2 and Exp_OCO3&2 are enhanced, by 0.08 and 0.05 PgC yr$^{-1}$, respectively.

519         On a global scale, the BIAS between the prior $CO_2$ concentrations and surface flask observations

is -1.82 ppm, with a MAE of 3.27 ppm and a RMSE of 5.01 ppm. The deviations between the posterior
$CO_2$ concentrations and surface flask observations for all three inversions are reduced to different de-
grees from the prior, especially for the BIAS, which decreased to 0.32, 0.02, and 0.05 ppm by
Exp_OCO3, Exp_OCO2, and Exp_OCO3&2, respectively. The reasons for the poor performance of
assimilating OCO-3 $XCO_2$ alone are, on the one hand, the fact that it is only available between 52° S
and 52°N, which leads to a lack of observational constraints on the carbon sinks at high latitudes, and
the large fluctuations in the amount of observational data, which leads to significant differences in
observational constraints at mid-latitudes at different times; on the other hand, its varied observation
time also affect the inversions, but even choosing afternoon observations does not improve the inver-
sions because the amount of observed data drops significantly. Therefore, a better option for the future
would be to jointly assimilate the OCO-2 $XCO_2$ data and the OCO-3 $XCO_2$ retrievals observed in the
afternoon (12:00 to 16:00 LST).

**Code availability.** The code of the GCASv2 system is available to the community and can be accessed
upon request from Fei Jiang(jiangf@nju.edu.cn) at Nanjing University.
**Data availability.** The OCO-2 and OCO-3 data used in this study is available at https://ww
w.earthdata.nasa.gov. The FOSSIL carbon emissions of GCP-GridFEDv2023.1 is available at
https://doi.org/10.5281/zenodo.8386803. The FIRE carbon emissions GFED 4.1s is available at
https://daac.ornl.gov/VEGETATION/guides/fire_emissions_v4_R1.html. The results of three in
version experiments and evaluation are publicly available at https://doi.org/10.5281/zenodo.112

540     39535.


**Author contributions.** XW and FJ designed the research. XW ran the model, analyzed the results
and wrote the paper. HW and ZZ collected the OCO-2 and OCO-3 $XCO_2$ retrievals. MW, JW, WH,
WJ and JC participated in the discussion of the inversion results and provided revisions before the
paper was submitted.

**Competing interests.** The author has declared that none of the authors has any competing interests.

**Financial support.** This work is supported by the National Key R&D Program of China (Grant No:
2023YFB3907404), the National Natural Science Foundation of China (Grant No. 42377102), and the
Fengyun Application Pioneering Project (Grant No: FY-APP-2022.0505).

**Acknowledgments.** The OCO-2 and OCO-3 data are produced by the OCO project at the Jet Propul-
sion Laboratory, California Institute of Technology, and obtained from the data archive at the NASA
Goddard Earth Science Data and Information Services Center. We acknowledge all atmospheric data
providers to obspack_co2_1_GLOBALVIEWplus_v9.1_2023-12-08. We are also grateful to the
High-Performance Computing Center (HPCC) of Nanjing University for doing the numerical calcu-
lations in this paper on its blade cluster system.

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
