# Peer review of "The role of OCO-3 XCO₂ retrievals in estimating global terrestrial net ecosystem exchanges"

_EGUsphere, 2024_

## Referee Comment (RC1)

The manuscript by *Wang et al.* (2024) analyzes global and regional terrestrial and oceanic carbon dioxide ($CO_2$) sources and sinks using OCO-2 and OCO-3 retrievals and the GCASv2 inverse modeling system. The purpose of this study was to determine the impact of using OCO-3 alone, and in conjunction with OCO-2, to estimate the global $CO_2$ budget. The results of this work show that there are large differences in $CO_2$ terrestrial fluxes in certain regions when comparing inverse model results when assimilating OCO-2 and OCO-3 separately. Also, when assimilating OCO-2+OCO-3 data it was determined that the results were similar to OCO-2 only model runs. When compared to estimated global atmospheric $CO_2$ growth rates and independent observations, it found that OCO-3 inversion results had larger biases/errors compared to OCO-2 only and OCO-2+OCO-3 simulations. The primary reason for this was attributed to the limited observational coverage of OCO-3 which does not observe $CO_2$ in the high latitudes. Overall, the paper is interesting and studies an important topic for using satellite-derived $CO_2$ for estimating global and regional $CO_2$ flux budget. The manuscript is generally well-written; however, lacks necessary information and context in many parts of the paper. After careful consideration of the minor and major comments provided below, I would expect this paper to be sufficient for publication.

**Minor Comments**

1. Line 56. Use "%" instead of "per cent".

2. Line 75. "The OCO satellites".

3. Line 77. Remove "a".

4. Line 91. "till" should be "until".

5. Line 114. Two-layer.

6. Prior emissions data. Do all 4 prior emissions data sets cover the entire time period of the model simulations (2019-2022)?

7. Line 195-196. Do the authors use GEOS-FP meteorological data? Please be clearer about the data used in the simulations.

8. Figure 1. Do the OCO-3 retrievals help with the lack of space-based XCO2 observations in the tropics compared to OCO-2? This would be very helpful to discuss.

9. Figure 3 c, f, i. It is very challenging to see much information from these subpanels.

10. Line 222. Did you mean to say "sources" here?

11. Line 334-336. This sentence is confusing. Why is the NEE from 2019 being discussed since the inversion was for 2020-2022? Also, where in Table 3 is it shown that the prior NEE is 3.5 PgC less than the posterior fluxes? I am guessing the authors might mean ppb instead of PgC?

**Major Comments**

1. Line 114-116. Can the authors please expand upon this two-layer localization scale used to filter observations to be used in the inversion? It's not clear from this sentence what is actually being done and why.

2. Section 2.1. The authors need to provide more information about the GCASv2 model. What is the horizontal spatial resolution of the system? What meteorological data is used to drive the simulations? How are prior emission and observation error covariance matrices developed? What are the main upgrades in GCASv2 compared to GCASv1? There is a lot of detail that could be added to this section in order for the reader to better understand the inversion system.

3. Use of OCO-2/3 Ocean Glint (OG) observations. The vast majority of research that assimilates OCO-2 and OCO-3 XCO2 data tend to avoid the usage of OG retrievals (e.g., Peiro et al., 2022; Byrne et al., 2023). This even includes studies which focus on constraining oceanic fluxes of $CO_2$ applying the newest version 11 OCO-2 retrievals (e.g., Jin et al., 2024). The reason for this is the OG retrievals from the OCO satellite sensors have been determined to potentially have unrealized errors/biases and spurious trends (Peiro et al., 2022; Byrne et al., 2023). I would suggest the authors run additional inversions only using land nadir and land glint (LN+LG) OCO-2 and OCO-3 retrievals to see how this impacts the results of this study. If there are noticeable differences, which is expected, the authors should update the results of this paper using LN+LG observations only or discuss the impacts of using OG retrievals on the results of the paper.

4. Model simulation spin-up and spin-down. The authors run their inverse model between August 2019 and December 2022 using a 5-month spin-up time. Did you allow for any spin-down time as well? Observations into early 2023 will still impact the inversion of regional/global $CO_2$ in 2022. It is common to apply multiple months of spin-up and spin-down when constraining global $CO_2$ fluxes with OCO-2/3 retrievals. The authors need to explain whether they provided spin-down months in their simulations and if not, consider running the simulation with at least 5 months (same as the spin-up time) of spin-down.

5. Sect. 3. Overall, more detail is needed about your inversion set up. For instance, what are the prior errors you apply for both terrestrial (NEE) and oceanic fluxes? This is extremely important for the results of this study. From Table 1 it appears ocean fluxes did not deviate too far from the prior estimates which leads me to assume the prior errors for these sources were low. What inversion method do you use? There is a glaring lack of information in this section.

6. Line 209. How was the average atmospheric $CO_2$ growth rate of 4.96 PgC yr-1 for 2020-2022 calculated from Friedlingstein et al. (2023)? Is the growth rate for each year provided in this report? I see values provided for 2022 by itself, but don't see the 2020-2022 average. There is interannual

variability in the global $CO_2$ growth rate so how you calculated this value can impact the value you compared to growth rates from OCO-2, OCO-3, and OCO-2+OCO-3 simulations.

7. Table 2. If you consider prior emission error/uncertainty, how many of these regions have inversion fluxes which are statistically different (at least considering 1 sigma uncertainty) from the a priori estimate? Also, do the authors calculate/consider posterior emission estimate uncertainty? Some of these values might not be different to a statistically significant level. The authors should expand upon this.

8. Line 352-354. Are the biases in OCO-3 only simulations really significantly smaller than OCO-2 only retrievals? The error bars in Fig. 6 don't suggest they are different to a statistically significant degree. Same thing when discussing the OCO-3 biases being higher in the Northern Hemisphere. It would be nice if the author provided more explanation in this paragraph about why there are large differences in biases. Is it all due to observational coverage differences between OCO-2 and OCO-3?

9. Line 89, 124-125. I am not sure it is correct to say that OCO-3 has the same temporal resolution as OCO-2. OCO-2 is in a near-polar orbit and observes points at the same time of day. OCO-3 is on the ISS orbit which differs from this and allows OCO-3 to observe points at different times of day. How might the fact that OCO-3 observes times other than the standard 13:30 LT observations from OCO-2 impact the results of the inversion?

10. The conclusion/discussion section of the paper could be improved with further information on the importance of these findings. How should studies in the future use both OCO-2 and OCO-3 for estimating global $CO_2$ fluxes? What are the pros and cons of OCO-3 alone? How does this project expand our knowledge of the global carbon cycle? Some additional text to identify the novelty of this study compared to the vast amount of literature using OCO-2 for inferring global and regional $CO_2$ fluxes would be very helpful.

**References**

Byrne, B., Baker, D. F., Basu, S., Bertolacci, M., Bowman, K. W., Carroll, D., Chatterjee, A., Chevallier, F., Ciais, P., Cressie, N., Crisp, D., Crowell, S., Deng, F., Deng, Z., Deutscher, N. M., Dubey, M. K., Feng, S., García, O. E., Griffith, D. W. T., Herkommer, B., Hu, L., Jacobson, A. R., Janardanan, R., Jeong, S., Johnson, M. S., Jones, D. B. A., Kivi, R., Liu, J., Liu, Z., Maksyutov, S., Miller, J. B., Miller, S. M., Morino, I., Notholt, J., Oda, T., O'Dell, C. W., Oh, Y.-S., Ohyama, H., Patra, P. K., Peiro, H., Petri, C., Philip, S., Pollard, D. F., Poulter, B., Remaud, M., Schuh, A., Sha, M. K., Shiomi, K., Strong, K., Sweeney, C., Té, Y., Tian, H., Velazco, V. A., Vrekoussis, M., Warneke, T., Worden, J. R., Wunch, D., Yao, Y., Yun, J., Zammit-Mangion, A., and Zeng, N.: National $CO_2$ budgets (2015–2020) inferred from atmospheric $CO_2$ observations in support of the global stocktake, Earth Syst. Sci. Data, 15, 963–1004, https://doi.org/10.5194/essd-15-963-2023, 2023.

Friedlingstein, P., O'Sullivan, M., Jones, M. W., Andrew, R. M., Bakker, D. C. E., Hauck, J., Landschützer, P., Le Quéré, C., Luijkx, I. T., Peters, G. P., Peters, W., Pongratz, J.,

Schwingshackl, C., Sitch, S., Canadell, J. G., Ciais, P., Jackson, R. B., Alin, S. R., Anthoni, P., Barbero, L., Bates, N. R., Becker, M., Bellouin, N., Decharme, B., Bopp, L., Brasika, I. B. M., Cadule, P., Chamberlain, M. A., Chandra, N., Chau, T.-T.-T., Chevallier, F., Chini, L. P., Cronin, M., Dou, X., Enyo, K., Evans, W., Falk, S., Feely, R. A., Feng, L., Ford, D. J., Gasser, T., Ghattas, J., Gkritzalis, T., Grassi, G., Gregor, L., Gruber, N., Gürses, Ö., Harris, I., Hefner, M., Heinke, J., Houghton, R. A., Hurtt, G. C., Iida, Y., Ilyina, T., Jacobson, A. R., Jain, A., Jarníková, T., Jersild, A., Jiang, F., Jin, Z., Joos, F., Kato, E., Keeling, R. F., Kennedy, D., Klein Goldewijk, K., Knauer, J., Korsbakken, J. I., Körtzinger, A., Lan, X., Lefèvre, N., Li, H., Liu, J., Liu, Z., Ma, L., Marland, G., Mayot, N., McGuire, P. C., McKinley, G. A., Meyer, G., Morgan, E. J., Munro, D. R., Nakaoka, S.-I., Niwa, Y., O'Brien, K. M., Olsen, A., Omar, A. M., Ono, T., Paulsen, M., Pierrot, D., Pocock, K., Poulter, B., Powis, C. M., Rehder, G., Resplandy, L., Robertson, E., Rödenbeck, C., Rosan, T. M., Schwinger, J., Séférian, R., Smallman, T. L., Smith, S. M., Sospedra-Alfonso, R., Sun, Q., Sutton, A. J., Sweeney, C., Takao, S., Tans, P. P., Tian, H., Tilbrook, B., Tsujino, H., Tubiello, F., van der Werf, G. R., van Ooijen, E., Wanninkhof, R., Watanabe, M., Wimart-Rousseau, C., Yang, D., Yang, X., Yuan, W., Yue, X., Zaehle, S., Zeng, J., and Zheng, B.: Global Carbon Budget 2023, Earth Syst. Sci. Data, 15, 5301–5369, https://doi.org/10.5194/essd-15-5301-2023, 2023.

Jin, Z., Tian, X., Wang, Y., Zhang, H., Zhao, M., Wang, T., Ding, J., and Piao, S.: A global surface $CO_2$ flux dataset (2015–2022) inferred from OCO-2 retrievals using the GONGGA inversion system, Earth Syst. Sci. Data, 16, 2857–2876, https://doi.org/10.5194/essd-16-2857-2024, 2024.

Peiro, H., Crowell, S., Schuh, A., Baker, D. F., O'Dell, C., Jacobson, A. R., Chevallier, F., Liu, J., Eldering, A., Crisp, D., Deng, F., Weir, B., Basu, S., Johnson, M. S., Philip, S., and Baker, I.: Four years of global carbon cycle observed from the Orbiting Carbon Observatory 2 (OCO-2) version 9 and in situ data and comparison to OCO-2 version 7, Atmos. Chem. Phys., 22, 1097–1130, https://doi.org/10.5194/acp-22-1097-2022, 2022.

---

## Author Response (AR1)

**Referee #1**

We thank the anonymous reviewer for his/her thorough review and valuable comments, which will help us to express our results more clearly. We have made revisions based on the referee's suggestions and have responded to all comments point by point. The page and line numbers of all revisions are referenced to the revised manuscript. References related to the responses are listed in the end of this document.

General comments.

The manuscript by Wang et al. (2024) analyzes global and regional terrestrial and oceanic carbon dioxide ($CO_2$) sources and sinks using OCO-2 and OCO-3 retrievals and the GCASv2 inverse modeling system. The purpose of this study was to determine the impact of using OCO-3 alone, and in conjunction with OCO-2, to estimate the global $CO_2$ budget. The results of this work show that there are large differences in $CO_2$ terrestrial fluxes in certain regions when comparing inverse model results when assimilating OCO-2 and OCO-3 separately. Also, when assimilating OCO-2+OCO-3 data it was determined that the results were similar to OCO-2 only model runs. When compared to estimated global atmospheric $CO_2$ growth rates and independent observations, it found that OCO-3 inversion results had larger biases/errors compared to OCO-2 only and OCO-2+OCO-3 simulations. The primary reason for this was attributed to the limited observational coverage of OCO-3 which does not observe $CO_2$ in the high latitudes. Overall, the paper is interesting and studies an important topic for using satellite-derived $CO_2$ for estimating global and regional $CO_2$ flux budget. The manuscript is generally well-written; however, lacks necessary information and context in many parts of the paper. After careful consideration of the minor and major comments provided below, I would expect this paper to be sufficient for publication.

Minor Comments.

1. Line 56. Use "%" instead of "per cent".

**Response**: Thank you for this suggestion. We have changed "per cent" to "%" (see Line 56, Page 3).

2. Line 75. "The OCO satellites".

**Response**: Thank you for this suggestion. We have revised it (see Line 75, Page 3)..

3. Line 77. Remove "a".

**Response**: Thank you for this suggestion. We have removed "a" (see Line 77, Page 3).

4. Line 91. "till" should be "until".

**Response**: Thank you for this suggestion. We have changed "till" to "until" (see Line 91, Page 4).

5. Line 114. Two-layer.

**Response**: Thank you for this suggestion. We have revised it (see Line 152, Page 6).

6. Prior emissions data. Do all 4 prior emissions data sets cover the entire time period of the model simulations (2019-2022)?

**Response**: Yes, all 4 prior emissions data sets used in this study cover the entire time period of the model simulations (2019-2022). We have added a sentence of "All 4 prior fluxes cover the entire time period of this study (i.e., August 2019 to December 2022) and they were re-grided to a unified spatial resolution of 1º×1º before used in the GCASv2 system." at the end of the 1st paragraph in section 2.3 (see Lines 219-221, Page 10).

7. Line 195-196. Do the authors use GEOS-FP meteorological data? Please be clearer about the data used in the simulations.

**Response**: No! we use GEOS-5 Global Atmosphere Forcing Data (Tilmes, 2016) meteorolgical data to drive the MOZART-4, not GEOS-FP. GEOS-FP data are used for analyses and forecasts in real time using the latest validated Global Earth Observing System of Systems (GEOS), mainly for real-time weather forecasts, air quality forecasts and short-term environmental studies. We can download GEOS-5 Global Atmosphere Forcing Data from this website: https://rda.ucar.edu/datasets/d313000/.

8. Figure 1. Do the OCO-3 retrievals help with the lack of space-based $XCO_2$ observations in the tropics compared to OCO-2? This would be very helpful to discuss.

**Response**: Yes, the OCO-3 $XCO_2$ retrievals help with the lack of space-based $XCO_2$ observations in the tropics compared to OCO-2. As shown in Figure 5, we can find that the BIAS of Exp_OCO3&2 is smaller than Exp_OCO2 in the region from 30°S to 30°N. Meanwhile, the BIAS of Exp_OCO3&2 is also smaller than Exp_OCO2 in southern Africa, northern Africa and tropical Asia. It can prove that the OCO-3 retrievals can solve the problem of lack of space-based $XCO_2$ observations in the tropics to a certain extent.
The following sentences have been added in the revised manuscript (see Lines 420-423, Page 19): "We also find that the OCO-3 $XCO_2$ retrievals help with the lack of space-based $XCO_2$ observations in the tropics compared to OCO-2. The BIAS of Exp_OCO3&2 is smaller than Exp_OCO2 in the region from 30°S to 30°N. Meanwhile, the BIAS of Exp_OCO3&2 are also smaller than Exp_OCO2 in southern Africa, northern Africa and tropical Asia."

9. Figure 3 c, f, i. It is very challenging to see much information from these subpanels.

**Response**: Thank you for this suggestion. We have modified Figure 3 to make the right subpanels clearer, see Page 16 in the revised manuscript.

10. Line 222. Did you mean to say "sources" here?

**Response**: Thank you! It should be "sources" here, we have corrected this mistake (see Line 283, Page 12).

11. Line 334-336. This sentence is confusing. Why is the NEE from 2019 being discussed since the inversion was for 2020-2022? Also, where in Table 3 is it shown that the prior NEE is 3.5 PgC less than the posterior fluxes? I am guessing the authors might mean ppb instead of PgC?

**Response**: Many thanks for this suggestion. Our inversion experiments started in August 2019, so the NEE generated during the spin-up stage will have an impact on the subsequent results. The prior NEE in 2019 is on average 3.5 PgC less than the posterior NEE, which leads to a large negative bias in the prior $CO_2$ concentrations in 2020-2022 to compensate for the difference due to the 2019 NEE. To further explain this question, we have revised that sentence (see Lines 396-399, Page 18) as

follows: "The prior BIAS shows a pronounced negative bias, which can be attributed to the fact that the prior NEE in 2019 (generated by the spin-up stage) was, on average, approximately 3.5 PgC less than the posterior NEE. This part of the NEE will has an impact on the subsequent inversion."

Major Comments.

1. Line 114-116. Can the authors please expand upon this two-layer localization scale used to filter observations to be used in the inversion? It's not clear from this sentence what is actually being done and why.

**Response**: Thank you for this suggestion. A two-layer localization scheme was employed to filter the observations used in the inversion, mainly to reduce the effect of spurious correlations. The localization technique in this study is based on the correlation coefficient between the simulated concentration ensembles in each observation location and the perturbed fluxes in current model grids and their distances. The observations will be accepted for assimilation if the distance is less than 500 km and the correlation coefficient is greater than zero; if the distance is greater than or equal to 500 km and less than 3000 km and the relationship between the simulated concentration and the perturbed fluxes is significant ($p<0.05$), the observations will be accepted for assimilation. Otherwise, the observations will not be accepted for assimilation. The reason for this scheme is that considering the atmospheric horizontal diffusion, we believe that there must be a correlation between the fluxes in the current grid and the concentrations in the neighbouring grids, and therefore observations are accepted as long as this correlation is greater than zero. In contrast, at distant locations (>500 km), where the effect of atmospheric horizontal diffusion is essentially negligible, the relationship between source and receptor is mainly due to atmospheric transport, and in order to minimise spurious correlations we require that such correlations must be significant.

We have added details about the two-layer localization scale in the revised manuscript (see Lines 152-166, Pages 6-7) as follows:

"There are inevitably spurious correlations in the EnKF method, to reduce the effect of spurious correlations, a two-layer localization scale was adopted in GCASv2, which is used to select which observations can be used for the flux analysis for each grid. The localization technique is based on the correlation coefficient between the simulated XCO$_2$ ensembles ($XCO_{2,i}^m$) in each observation location and the perturbed fluxes ($X_i^b$) in current model grids and their distances. The observations will be accepted for assimilation if the distance is less than 500 km and the correlation coefficient is greater than 0; and if the distance is greater than or equal to 500 km and less than 3000 km and the correlation coefficient should be significant (p<0.05). Otherwise, the observations are not accepted. The reason for this scheme is that considering the atmospheric horizontal diffusion, we believe that there must be a correlation between the flux of one grid and the concentrations in its neighbouring grids, and therefore observations are accepted as long as this correlation coefficient is greater than zero. In contrast, at distant locations (>500 km), where the effect of atmospheric horizontal diffusion is essentially negligible, the relationship between source and receptor is mainly due to atmospheric transport, and in order to minimize spurious correlations we require that such correlations must be significant."

2. Section 2.1. The authors need to provide more information about the GCASv2 model. What is

the horizontal spatial resolution of the system? What meteorological data is used to drive the simulations? How are prior emission and observation error covariance matrices developed? What are the main upgrades in GCASv2 compared to GCASv1? There is a lot of detail that could be added to this section in order for the reader to better understand the inversion system.

**Response**: Thank you for this suggestion. We have added detailed description about the GCASv2 system and its settings in this study in the revised manuscript, including the major upgrades over GCASv1, the atmospheric transport model and its driver data, the resolutions, the data assimilation algorithms and core formulas, the detailed flow for each assimilation window, the "super-observation" scheme, the localization scheme, and the background and observation error covariance matrices.

In Section 2.1 (see Pages 4-7, lines 107-166),

[revised manuscript text omitted]

3. Use of OCO-2/3 Ocean Glint (OG) observations. The vast majority of research that assimilates OCO-2 and OCO-3 $XCO_2$ data tend to avoid the usage of OG retrievals (e.g., Peiro et al., 2022; Byrne et al., 2023). This even includes studies which focus on constraining oceanic fluxes of $CO_2$ applying the newest version 11 OCO-2 retrievals (e.g., Jin et al., 2024). The reason for this is the OG retrievals from the OCO satellite sensors have been determined to potentially have unrealized errors/biases and spurious trends (Peiro et al., 2022; Byrne et al., 2023). I would suggest the authors run additional inversions only using land nadir and land glint (LN+LG) OCO-2 and OCO-3 retrievals to see how this impacts the results of this study. If there are noticeable differences, which is expected, the authors should update the results of this paper using LN+LG observations only or discuss the impacts of using OG retrievals on the results of the paper.

**Response**: Thank you for this suggestion. Indeed, in most of the previous studies that used OCO-2 $XCO_2$ to invert surface carbon fluxes, the OG data were not used. The reason is that the OG $XCO_2$ may have larger uncertainties, inversions assimilating OCO-2 OG retrievals produced unrealistic results of annual global ocean sinks (Peiro et al., 2022). In addition to its large uncertainties, we believe that another reason for the poor assimilation performance of OG is the relatively homogeneous distribution of $XCO_2$ on ocean, causing a large correlation of the model-data biases among different $XCO_2$ observations within a same region, which leads to observations at the same region having the same direction of adjustment for surface fluxes, and thus leads to a significant overestimated or underestimated ocean carbon sink. Because of this, some assimilation algorithms (e.g., EnSRF) can only achieve better assimilation results when the model-data biases between observations have relatively small correlation or are uncorrelated. Therefore, in this study, we set the OG data with larger uncertainties than the LNLG data, and re-grided it at a coarser spatial resolution of 5°×5°. The results show that under this scheme, the inverted ocean sink is reasonable, with value of -2.6 PgC yr$^{-1}$ (Table 1). According to the reviewer's suggestion, we have added three additional inversion experiments in the revised manuscript, in which we use only land nadir and land glint (LN+LG) OCO-2 and OCO-3 retrievals for the inversion (Named as Exp_OCO3L, Exp_OCO2L and Exp_OCO3&2L, respectively). We compared the estimates of NEE and the evaluation against *in-situ* observations between the experiments with and without OG data, and found that assimilating OG data with our method can improve the inversions somewhat compared to removing OG.

We have added a paragraph in Section 4.5 in the revised manuscript (see Lines 439-462, Pages 20-21):

"In most of the previous studies that used OCO-2 $XCO_2$ to invert surface carbon fluxes, the OG data were not used (e.g., Peiro et al., 2022; Byrne et al., 2023), the reason is that the OG $XCO_2$ may have larger uncertainties, inversions assimilating OCO-2 OG retrievals produced unrealistic results of annual global ocean sinks (Peiro et al., 2022). In addition to its large uncertainties, we believe that another reason for the poor assimilation performance of OG is the relatively homogeneous distribution of $XCO_2$ on ocean, causing a large correlation of the model-data biases among different $XCO_2$ observations within a same region, which leads to observations at the same region having the same direction of adjustment for surface fluxes, and thus leads to a significant overestimated or underestimated of ocean carbon sink. Because of this, some assimilation algorithms (e.g., EnSRF) can only achieve better assimilation results when the model-data biases between observations have

relatively small correlation or are uncorrelated. Therefore, in this study, we set the OG data with larger uncertainties than the LNLG data, and re-grided it at a coarser spatial resolution of 5°×5°. The results show that under this scheme, the inverted ocean sink is reasonable, with value of -2.6 PgC yr$^{-1}$ (Table 1). In addition, in order to compare the scheme that we have adopted in this study with the previous scheme that do not assimilate the OG, we added three additional inversion experiments, in which only the LNLG data were assimilated (Table S1). It could be found that all the three inversion experiments without OG observations place smaller constraints on the ocean fluxes compared to the original experiments, with the posterior ocean fluxes remaining almost identical to the prior ocean fluxes. Correspondingly, the inverted global land sink as well as the sinks in most regions show a slight decrease (Tables S2 and S3). Evaluations in comparison with in-situ observations showed that there are some increases in the a posteriori concentration biases for all three experiments after removing OG. For example, for the experiments assimilating OCO-2 data, the mean bias increased from 0.02 to 0.14 ppm (Table S4). This suggests that assimilating OG data with our method can improve the inversions somewhat compared to removing OG."

4. Model simulation spin-up and spin-down. The authors run their inverse model between August 2019 and December 2022 using a 5-month spin-up time. Did you allow for any spin-down time as well? Observations into early 2023 will still impact the inversion of regional/global $CO_2$ in 2022. It is common to apply multiple months of spin-up and spin-down when constraining global $CO_2$ fluxes with OCO-2/3 retrievals. The authors need to explain whether they provided spin-down months in their simulations and if not, consider running the simulation with at least 5 months (same as the spin-up time) of spin-down.

**Response**: Thank you for this suggestion. The use of spin-up is mainly due to the fact that the initial field of concentration at the beginning of the inversion may have a large error, while spin-down is mainly due to the fact that the global transport of atmospheric $CO_2$ takes time, and later observations will also affect the current inversion results. In our system, we run our inverse model with a 5-month spin-up time, but for the spin-down, since the assimilation window is 1 week and the subsequent observations will not impact the inversion results of current DA window, so spin-down time is not required in current version of the GCAS system. The reason for adopting this scheme is, on the one hand, to take into account the fact that the amount of observation data from satellites has increased considerably compared to that from ground stations, and that within a one-week assimilation window, the included observation data can already provide a better constraint on the fluxes, and on the other hand, increasing the length of the window or adopting a sliding window, although it is possible to include observation data from farther away, the signals of atmospheric concentrations due to flux changes in a certain grid decay with time and distance, and observations at distant places can sense the signals but they are very weak, especially for $XCO_2$, while expanding the distance may include more spurious observation signals, which may make the inversion results worse.

5. Sect. 3. Overall, more detail is needed about your inversion set up. For instance, what are the prior errors you apply for both terrestrial (NEE) and oceanic fluxes? This is extremely important for the results of this study. From Table 1 it appears ocean fluxes did not deviate too far from the prior estimates which leads me to assume the prior errors for these sources were low. What inversion method do you use? There is a glaring lack of information in this section.

**Response**: Thank you for this suggestion. Indeed, the adjustments to ocean carbon fluxes in this study are relatively small. On the one hand, although the global ocean carbon sink is comparable to that of the global land, the intensity of the carbon flux per unit area of the oceans is actually much smaller than that of the land mass because the oceans are much larger than the land mass. As a result, the signals of changes in ocean carbon fluxes over the same area that can be sensed by satellite observations of $XCO_2$ are indeed weaker. On the other hand, we have processed the ocean OG data to increase its uncertainty and re-grided it to a $5° \times 5°$ resolution, which further weakens the constraints on the ocean carbon flux from satellite $XCO_2$. We have added more information about the inversion set up in the revised manuscript.

In Section 2.1 (See Lines 121-124, Page 5):

"……the prior fluxes ($X_0^b$) in each grid are independently perturbed with a random number ($\delta_i$) drawn from a Gaussian distribution with mean of 0 and standard deviation of 1, and a scaling factor ($\lambda$) that represents the uncertainty of each prior flux (Eq. 2).

$$X_i^b = X_0^b + \lambda \times \delta_i \times X_0^b \quad , i = 1, 2, ... , N \qquad (1)"$$

In Section 3 (See Lines 245-254, Page 11):

"……According to Eq. (1), the prior NEE and OCN fluxes were perturbed using Eq. (3).

$$X_i^b = \lambda_{NEE} \times \delta_{i,NEE} \times X_{NEE}^b + \lambda_{ocn} \times \delta_{i,ocn} \times X_{OCN}^b + X_{Fire}^b + X_{Fossil}^b, \, i = 1, 2, ..., N \quad (3)$$

where $X_{NEE}^b$, $X_{OCN}^b$, $X_{Fire}^b$, and $X_{Fossil}^b$ represent the prior fluxes of NEE, OCN, FIRE, and FOSSIL, respectively; $\delta_i$ is random perturbation samples, which is independent between grids; $\lambda_{NEE}$ and $\lambda_{ocn}$ are the scaling factors for prior NEE and OCN fluxes, which were set to be 6 and 10 in this study, respectively."

6. Line 209. How was the average atmospheric $CO_2$ growth rate of 4.96 PgC yr-1 for 2020-2022 calculated from Friedlingstein et al. (2023)? Is the growth rate for each year provided in this report? I see values provided for 2022 by itself, but don't see the 2020-2022 average. There is interannual variability in the global $CO_2$ growth rate so how you calculated this value can impact the value you compared to growth rates from OCO-2, OCO-3, and OCO-2+OCO-3 simulations.

**Response**: Thank you very much for pointing this out, when we conducted the inversion work, GCB2023 (i.e., Friedlingstein et al., 2023) had not yet been released, we used the 2020 and 2021 data (4.99 and 5.23 PgC/yr) from GCB2022, as well as the Annual Mean Global Carbon Dioxide Growth Rates (2.2 ppm) in 2022 reported by NOAA Global Monitoring Laboratory (https://gml.noaa.gov/ccgg/trends/gl_gr.html) by multi-by a factor of 2.124. The average atmospheric $CO_2$ growth rate is 4.96 PgC yr[-1] for 2020-2022. We compared the results in GCB2022 and GCB2023 and found there are some differences in these values. In GCB2023, the $CO_2$ growth rates from 2020 to 2022 have been updated to 4.97016, 5.2038, and 4.63032 PgC/yr, with mean of 4.93 PgC/yr. Therefore, in the revised manuscript, we have updated this value to 4.93 PgC/yr.

7. Table 2. If you consider prior emission error/uncertainty, how many of these regions have inversion fluxes which are statistically different (at least considering 1 sigma uncertainty) from the a priori estimate? Also, do the authors calculate/consider posterior emission estimate uncertainty?

Some of these values might not be different to a statistically significant level. The authors should expand upon this.

**Response**: Thank you for this suggestion. We have modified Table 1 and Table 2 to add uncertainties for globe and each Transcom-3 region. We conducted a z-test to investigate whether the inversion fluxes are statistically different the prior fluxes. Our analysis revealed in Europe and tropical S. America, the differences between the posterior and prior fluxes are significant. Therefore, in the revised manuscript, we used 'significant' only for describing the differences in these two regions (see Lines 320-327, Page 14). In addition, although we cannot directly see a significant difference between the prior and posterior annual fluxes in the other regions, we can find that in most regions, the differences between the monthly prior NEE and the posterior NEE of the three inversion experiments are significant, especially in the summer months (Figure 4 in the revised manuscript).

8. Line 352-354. Are the biases in OCO-3 only simulations really significantly smaller than OCO-2 only retrievals? The error bars in Fig. 6 don't suggest they are different to a statistically significant degree. Same thing when discussing the OCO-3 biases being higher in the Northern Hemisphere. It would be nice if the author provided more explanation in this paragraph about why there are large differences in biases. Is it all due to observational coverage differences between OCO-2 and OCO-3?

**Response**: Thank you for this suggestion. The deviations for each latitudinal zones in Fig. 5 are calculated by averaging the deviations of all surface stations located in that latitudinal band, where the error bars are the standard deviation of the average biases of each station. We conduct a t-test and found that there is a statistically significant difference between the biases in Exp_OCO3 and Exp_OCO2 only in the south of 60°S and north of 60°N ($p<0.05$). So, we removed the word 'significant' in the revised manuscript (See lines 414 and 417, page 19).

In the region north of 60°N, Exp_OCO3 exhibits a significant positive deviation, which is actually significantly different from that of Exp_OCO2 and Exp_OCO3&2 ($p<0.05$). Possible reasons for the poor performance of assimilating OCO-3 $XCO_2$ (Exp_OCO3) at high latitudes are 1) OCO-3 lacks observations beyond 52° North and South latitudes (Figure 1a); 2) the observation time different from OCO-2; and 3) its spatial coverage between 52°S and 52°N. We first examined weekly changes in the data amount of OCO-3 using the re-grided data as described in Section 2.3, and found that there are very significant cyclical fluctuations in the data amount from OCO-3 (Figure S4a). For the observation time, all observations of OCO-2 were at 1:30 p.m. local time (LST), whereas that of OCO-3 were variable, with only about 14% of the observations near 13:30 p.m. LST and about 54% in the morning or after 4:00 p.m. LST (Figure S1). In order to quantify these effects, we added another 3 additional inversion experiments, which were named as Exp_OCO2r, Exp_OCO3tc, and Exp_OCO2ts (Table S1). In Exp_OCO2r, only the OCO-2 $XCO_2$ retrievals located between 52°S and 52°N retrievals were assimilated, in Exp_OCO3tc, all the observation times of the OCO-3 XCO2 retrievals were changed to 1.30 p.m. LST, and in Exp_OCO3ts, only OCO-3 data with observation times between 12 and 3 p.m. LST were assimilated. We find that the lack of data beyond 52° North and South latitudes is the main reason for the poor assimilation of OCO-3, and the observation time as well as the cyclical variations in the observation number also have an important effect on the results. In the revised manuscript, we have added two long paragraphs to discuss the issue.

We have added the following paragraphs in Section 4.5 in the revised manuscript (see Lines 463-

"Since OCO-3 has similar observation uncertainties of $XCO_2$ with OCO-2 (Taylor et al., 2023), the poor performance of assimilating OCO-3 $XCO_2$ retrievals (Exp_OCO3) may be related to that 1) OCO-3 lacks observations beyond 52° North and South latitudes (Figure 1a); 2) the observation time different from OCO-2; and 3) its spatial coverage between 52°S and 52°N. We first examined weekly changes in the data amount of OCO-3 using the re-grided data as described in Section 2.3, and found that there are very significant cyclical fluctuations in the data amount from OCO-3 (Figure S4a). Every 8 weeks or so, there is a trough in the data amount. There is a difference of about 5 times between the weeks with the highest and the lowest data amount, and in the weeks with least data amount, there were essentially no observations in the northern hemisphere (Figure S4b). This implies that the surface carbon fluxes are largely unconstrained in the Northern Hemisphere, especially at mid- to high-latitudes, during the weeks with low observational data, resulting in poorer assimilation performance than for OCO-2. For the observation time, all observations of OCO-2 were at 1:30 p.m. local time (LST), whereas that of OCO-3 were variable, with only about 14% of the observations near 13:30 p.m. LST and about 54% in the morning or after 4:00 p.m. LST (Figure S1). For reasons such as coarser model resolution, the global atmospheric chemical transport models generally simulate atmospheric concentrations better only in the afternoon, when boundary layer heights are at their highest and atmospheric mixing is at its best, so assimilating these observations in the morning and after 4 p.m. LST may result in poorer inversions due to the greater simulation bias of the atmospheric transport models at these times of day.

In order to quantify these effects, we added another 3 additional inversion experiments, which were named as Exp_OCO2r, Exp_OCO3tc, and Exp_OCO2ts (Table S1). In Exp_OCO2r, only the OCO-2 $XCO_2$ retrievals located between 52°S and 52°N retrievals were assimilated, in Exp_OCO3tc, all the observation times of the OCO-3 $XCO_2$ retrievals were changed to 1.30 p.m. LST, and in Exp_OCO3ts, only OCO-3 data with observation times between 12 and 3 p.m. LST were assimilated. When the OCO-2 data beyond 52° North and South latitudes were also removed (Exp_OCO2r), the NEE estimates, both globally and for individual regions, are close to those of the Exp_OCO3 experiment, especially in the high latitude region of Europe and boreal North America, the inverted NEEs are almost identical to those of the Exp_OCO3 experiment (Table S2 and S3), and the bias of a posteriori concentrations from observations at high latitudes is close to that of the OCO-3 experiment (Figure S3). However, globally, compared to the OCO-3 experiment, the Exp_OCO2r experiment still has smaller the deviation between the global net flux and the observed annual growth rate (Table S2), and smaller the global mean bias of the posterior concentrations (Table S4). This suggests that the lack of observations of OCO-3 beyond 52° North and South latitudes does have a significant impact on the inversion results. In addition, it can also be noted that at mid-latitudes, the bias of Exp_OCO2r is also smaller than the OCO-3 experiment, which may be caused by the significant fluctuations in the data amount of OCO-3 (Figure S4). When we changed all the observation times of the OCO-3 $XCO_2$ retrievals to 1.30 p.m. LST (Exp_OCO3tc), although we are not actually able to do so, the inversion does show a significant improvement compared to Exp_OCO3. However, if we only select the data with observation time between 12:00 and 3:00 p.m. LST (Exp_OCO3ts), the deviation between the global net flux and the observed annual growth rate, and the mean biases of the posterior concentrations at most latitudes are larger than those of Exp_OCO3 (Table S2 and Figure S3), indicating a poorer performance than Exp_OCO3. The probably reason is that the data number of observations is substantially reduced at this time (Figure

S2), which leads to a substantial weakening of the observational constraints on surface carbon fluxes (Figure S5)."

9. Line 89, 124-125. I am not sure it is correct to say that OCO-3 has the same temporal resolution as OCO-2. OCO-2 is in a near-polar orbit and observes points at the same time of day. OCO-3 is on the ISS orbit which differs from this and allows OCO-3 to observe points at different times of day. How might the fact that OCO-3 observes times other than the standard 13:30 LT observations from OCO-2 impact the results of the inversion?

**Response**: Many thanks for this suggestion. Indeed, the time interval of the OCO-3 $XCO_2$ is different from that of OCO-2, OCO-3 has different observation times on different dates for the same place. In order to explore the impact of the observation times on the inversion results, we added another 2 additional inversion experiments, which were named as Exp_OCO3tc, and Exp_OCO2ts. In Exp_OCO3tc, all the observation times of the OCO-3 $XCO_2$ retrievals were changed to 1.30 p.m. LST, and in Exp_OCO3ts, only OCO-3 data with observation times between 12 and 3 p.m. LST were assimilated. In the Exp_OCO3tc, it performed better than Exp_OCO3. However, if we select only the OCO-3 observations with their observation time between 12:00 p.m. and 3:00 p.m. local time, the deviation between the global net flux and the observed annual growth rate, and the mean biases of the posterior concentrations at most latitudinal zones are larger than those of Exp_OCO3 (Table S2 and Figure S3).

We have added this result in the revised manuscript as follows:

Lines 474-481, Page 22:

"For the observation time, all observations of OCO-2 were at 1:30 p.m. local time (LST), whereas that of OCO-3 were variable, with only about 14% of the observations near 13:30 p.m. LST and about 54% in the morning or after 4:00 p.m. LST (Figure S1). For reasons such as coarser model resolution, the global atmospheric chemical transport models generally simulate atmospheric concentrations better only in the afternoon, when boundary layer heights are at their highest and atmospheric mixing is at its best, so assimilating these observations in the morning and after 4 p.m. LST may result in poorer inversions due to the greater simulation bias of the atmospheric transport models at these times of day."

Lines 498-506, Page 23:

"When we changed all the observation times of the OCO-3 $XCO_2$ retrievals to 1.30 p.m. LST (Exp_OCO3tc), although we are not actually able to do so, the inversion does show a significant improvement compared to Exp_OCO3. However, if we only select the data with observation time between 12:00 and 3:00 p.m. LST (Exp_OCO3ts), the deviation between the global net flux and the observed annual growth rate, and the mean biases of the posterior concentrations at most latitudes are larger than those of Exp_OCO3 (Table S2 and Figure S3), indicating a poorer performance than Exp_OCO3. The probably reason is that the data number of observations is substantially reduced at this time (Figure S2), which leads to a substantial weakening of the observational constraints on surface carbon fluxes (Figure S5)."

10. The conclusion/discussion section of the paper could be improved with further information on the importance of these findings. How should studies in the future use both OCO-2 and OCO-3 for estimating global $CO_2$ fluxes? What are the pros and cons of OCO-3 alone? How does this project

expand our knowledge of the global carbon cycle? Some additional text to identify the novelty of this study compared to the vast amount of literature using OCO-2 for inferring global and regional $CO_2$ fluxes would be very helpful.

**Response**: Thank you for this suggestion. The OCO-3 satellite observations have a sufficient number of observations in the mid-latitude land region, while the OCO-2 satellite observations have a wide spatial coverage, even at high latitudes (Figure 1 in the original manuscript). Therefore, Exp_OCO3&2 assimilates sufficient observations in the mid-latitude region and observations in the high-latitude region, and has the advantages of OCO-2 and OCO-3 at the same time. At the same time, the joint assimilation of OCO-2 and OCO-3 $XCO_2$ also absorbs more observations than assimilating the OCO-2 or OCO-3 alone, which will also make the assimilation better. The advantage of OCO-3 satellite observations is that it has a sufficient number of observations in the mid-latitude land region, so assimilating the OCO-3 will perform better in optimizing $CO_2$ fluxes over mid-latitude landmasses. But it lacked the observation of high latitudes, so it will perform worse to be there. Exp_OCO3tc has shown that assimilating OCO-3 observations at 12-3 p.m. LST improves assimilation, and therefore the time of observation should also be a condition for screening satellite data when using satellite observations to invert global carbon fluxes.

We have enriched our section 5 (see Lines 534-542, Page 24) as follows:

"The reasons for the poor performance of assimilating OCO-3 $XCO_2$ alone are, on the one hand, the fact that it is only available between 52°S and 52°N, which leads to a lack of observational constraints on the carbon sinks at high latitudes, and the large fluctuations in the amount of observational data, which leads to significant differences in observational constraints at mid-latitudes at different times; on the other hand, its varied observation time also affect the inversions, but even choosing afternoon observations does not improve the inversions because the amount of observed data drops significantly. Therefore, a better option for the future would be to jointly assimilate the OCO-2 $XCO_2$ data and the OCO-3 $XCO_2$ retrievals observed in the afternoon (12:00 to 16:00 LST)."


**Referee #2**

We would like to thank the anonymous referee for his/her comprehensive review and valuable suggestions. We have made revisions based on the referee's suggestions and have responded to all comments point by point. The page and line numbers of all revisions are referenced to the revised manuscript. References related to the responses are listed in the end of this document.

Summary:

In this work, the authors conduct atmospheric $CO_2$ inversions to estimate global NEEs using OCO-2 and OCO-3 $XCO_2$ retrievals and the Global Carbon Assimilation System, version 2(GCASv2). Three sets of experiments have been designed by the authors to evaluate the impact of using different OCO $XCO_2$ observations to constrain the posterior carbon fluxes: using OCO-3 XCO_2 only; using OCO-2 $XCO_2$ only; and using OCO-2 & OCO-3 $XCO_2$ combined. The overall results suggest using combined OCO-2&OCO-3 $XCO_2$ retrievals can yield better consistency when compared with in-situ observations while using OCO-3 $XCO_2$ retrievals alone presents largest biases. The results and discussion reveal some interesting patterns in global and regional NEEs across different experimental setups and provided some insights on the choice of satellite observations to constrain global NEEs, but lack in-depth discussion of the resulted behavior of using OCO-3 $XCO_2$ only, OCO-2 $XCO_2$ only, and OCO-2&OCO-3 $XCO_2$ combined. Please see below sections for detailed comments and I would expect the manuscript to be published once comments and questions have been resolved.

Main comments/questions:

More information needed for the GCASv2. I understand the GCASv2 is an established model and described in at least two other published journal articles, but detailed information on the model setup, inversion methods, and error covariance metrics can be very helpful for readers of this manuscript to better understand the inversion system and results. Also see related comments in the Technical notes section.

**Response**: Thank you for this suggestion. We have added more information about GCASv2 in section 2.1 and the inversion settings in this study in section 3 in the revised manuscript.

In Section 2.1, the following paragraphs or sentences have been added (Pages 4-7, lines 107-166):

[revised manuscript text omitted]

How are the posterior fluxes constrained when there's no observation data in GCASv2? For example, in the Exp_OCO3 at high latitudes, I would assume the posterior fluxes are less updated and would be similar to prior fluxes since no new information has been presented to the inversion system, but Figure 3 and Figure 6 seem to suggest the posterior fluxes changed substantially when compared to prior. More information on the EnSRF would be helpful for the readers to understand the inversion process.

**Response**: Thank you for this suggestion. Since the atmosphere is moving, a change in flux at a certain location can cause a change in concentration downwind, i.e., observations downwind can sense the flux change at that location, and thus we can use observations downwind to constrain the flux in that area. At high latitudes, although there are no observations of the OCO-3, observations downwind this region will be absorbed for assimilation by two-layer localization technique. The two-layer localization was employed to filter the observations used in the inversion, mainly to reduce the effect of spurious correlations. In the revised manuscript, we have added more information about the EnSRF and the two-layer localization technique, which has been detailed in the content of the previous response.

I'm curious about the authors' insights on why in general using OCO-2 XCO$_2$ alone and OCO-2&OCO-3 XCO$_2$ combined outperforms the experiment using OCO-3 only? Would there be any other reason except the spatial coverage and potential bias in OCO-3 XCO$_2$(line 263)?

**Response**: Thanks! We further analyzed the reasons for the poor assimilation of OCO-3 XCO$_2$ alone, and found that, in addition to the absence of observations in regions beyond 52° North and South latitudes, the varied observations timing and the cyclical variations in the observation data amount had an important influence on the inversion results. We first examined weekly changes in the data amount of OCO-3 using the re-grided data as described in Section 2.3, and found that there are very

significant cyclical fluctuations in the data amount from OCO-3 (Figure S4a). For the observation time, all observations of OCO-2 were at 1:30 p.m. local time (LST), whereas that of OCO-3 were variable, with only about 14% of the observations near 13:30 p.m. LST and about 54% in the morning or after 4:00 p.m. LST (Figure S1). In order to quantify these effects, we added 3 additional inversion experiments, which were named as Exp_OCO2r, Exp_OCO3tc, and Exp_OCO2ts (Table S1). In Exp_OCO2r, only the OCO-2 XCO$_2$ retrievals located between 52°S and 52°N retrievals were assimilated, in Exp_OCO3tc, all the observation times of the OCO-3 XCO$_2$ retrievals were changed to 1.30 p.m. LST, and in Exp_OCO3ts, only OCO-3 data with observation times between 12 and 3 p.m. LST were assimilated. We find that the lack of data beyond 52° North and South latitudes is the main reason for the poor assimilation of OCO-3, and the observation time as well as the cyclical variations in the observation number also have an important effect on the results. In the revised manuscript, we have added two long paragraphs to discuss the issue.

We have added the following paragraphs in Section 4.5 in the revised manuscript (see Lines 463-506, Pages 21-23):

"Since OCO-3 has similar observation uncertainties of XCO$_2$ with OCO-2 (Taylor et al., 2023), the poor performance of assimilating OCO-3 XCO$_2$ retrievals (Exp_OCO3) may be related to that 1) OCO-3 lacks observations beyond 52° North and South latitudes (Figure 1a); 2) the observation time different from OCO-2; and 3) its spatial coverage between 52°S and 52°N. We first examined weekly changes in the data amount of OCO-3 using the re-grided data as described in Section 2.3, and found that there are very significant cyclical fluctuations in the data amount from OCO-3 (Figure S4a). Every 8 weeks or so, there is a trough in the data amount. There is a difference of about 5 times between the weeks with the highest and the lowest data amount, and in the weeks with least data amount, there were essentially no observations in the northern hemisphere (Figure S4b). This implies that the surface carbon fluxes are largely unconstrained in the Northern Hemisphere, especially at mid- to high-latitudes, during the weeks with low observational data, resulting in poorer assimilation performance than for OCO-2. For the observation time, all observations of OCO-2 were at 1:30 p.m. local time (LST), whereas that of OCO-3 were variable, with only about 14% of the observations near 13:30 p.m. LST and about 54% in the morning or after 4:00 p.m. LST (Figure S1). For reasons such as coarser model resolution, the global atmospheric chemical transport models generally simulate atmospheric concentrations better only in the afternoon, when boundary layer heights are at their highest and atmospheric mixing is at its best, so assimilating these observations in the morning and after 4 p.m. LST may result in poorer inversions due to the greater simulation bias of the atmospheric transport models at these times of day.

In order to quantify these effects, we added another 3 additional inversion experiments, which were named as Exp_OCO2r, Exp_OCO3tc, and Exp_OCO2ts (Table S1). In Exp_OCO2r, only the OCO-2 XCO$_2$ retrievals located between 52°S and 52°N retrievals were assimilated, in Exp_OCO3tc, all the observation times of the OCO-3 XCO$_2$ retrievals were changed to 1.30 p.m. LST, and in Exp_OCO3ts, only OCO-3 data with observation times between 12 and 3 p.m. LST were assimilated. When the OCO-2 data beyond 52° North and South latitudes were also removed (Exp_OCO2r), the NEE estimates, both globally and for individual regions, are close to those of the Exp_OCO3 experiment, especially in the high latitude region of Europe and boreal North America, the inverted NEEs are almost identical to those of the Exp_OCO3 experiment (Table S2 and S3), and the bias of a posteriori concentrations from observations at high latitudes is close to that of the

OCO-3 experiment (Figure S3). However, globally, compared to the OCO-3 experiment, the Exp_OCO2r experiment still has smaller the deviation between the global net flux and the observed annual growth rate (Table S2), and smaller the global mean bias of the posterior concentrations (Table S4). This suggests that the lack of observations of OCO-3 beyond 52° North and South latitudes does have a significant impact on the inversion results. In addition, it can also be noted that at mid-latitudes, the bias of Exp_OCO2r is also smaller than the OCO-3 experiment, which may be caused by the significant fluctuations in the data amount of OCO-3 (Figure S4). When we changed all the observation times of the OCO-3 $XCO_2$ retrievals to 1.30 p.m. LST (Exp_OCO3tc), although we are not actually able to do so, the inversion does show a significant improvement compared to Exp_OCO3. However, if we only select the data with observation time between 12:00 and 3:00 p.m. LST (Exp_OCO3ts), the deviation between the global net flux and the observed annual growth rate, and the mean biases of the posterior concentrations at most latitudes are larger than those of Exp_OCO3 (Table S2 and Figure S3), indicating a poorer performance than Exp_OCO3. The probably reason is that the data number of observations is substantially reduced at this time (Figure S2), which leads to a substantial weakening of the observational constraints on surface carbon fluxes (Figure S5). ”

General comments:

Line 113: How does GCASv2 handle parameters of the aggregated 'super-observation'? For example, if multiple OCO soundings has been aggregated into one 'super-observation', how does GCASv2 incorporate information such as pressure weighting function and averaging kernels from each individual soundings?

**Response**: Thank you for this suggestion. The 'super-observations' are generated by averaging all observations within an assimilation window for the same model grid. In this method, it first calculates the simulated $XCO_2$ corresponding to each observed $XCO_2$ based on the observation time and location, and then, it performs a retrieval error-weighted average for all the simulated and observed $XCO_2$ falling within the same model grid in the DA window, respectively. In the revised manuscript, we have added the following sentence (Lines 148-151, Page 6) to make it clear.

"……a single high-precision "super-observation". In this method, it first calculates the simulated $XCO_2$ corresponding to each observed $XCO_2$ based on the observation time and location, and then, it performs a retrieval error-weighted average for all the simulated and observed $XCO_2$ falling within the same model grid in the DA window, respectively."

Line 131: Can you justify the use of ocean glint? Ocean glint data is in general avoided in inversions due to potential high bias.

**Response**: Thank you for this suggestion. Indeed, in most of the previous studies that used OCO-2 $XCO_2$ to invert surface carbon fluxes, the OG data were not used. The reason is that the OG $XCO_2$ may have larger uncertainties, inversions assimilating OCO-2 OG retrievals produced unrealistic results of annual global ocean sinks (Peiro et al., 2022). In addition to its large uncertainties, we believe that another reason for the poor assimilation performance of OG is the relatively homogeneous distribution of $XCO_2$ on ocean, causing a large correlation of the model-data biases among different $XCO_2$ observations within a same region, which leads to observations at the same

region having the same direction of adjustment for surface fluxes, and thus leads to a significant overestimated or underestimated ocean carbon sink. Because of this, some assimilation algorithms (e.g., EnSRF) can only achieve better assimilation results when the model-data biases between observations have relatively small correlation or are uncorrelated. Therefore, in this study, we set the OG data with larger uncertainties than the LNLG data, and re-grided it at a coarser spatial resolution of 5°×5°. The results show that under this scheme, the inverted ocean sink is reasonable, with value of -2.6 PgC yr$^{-1}$ (Table 1). According to the reviewer 1's suggestion, we have added three additional inversion experiments in the revised manuscript, in which we use only land nadir and land glint (LN+LG) OCO-2 and OCO-3 retrievals for the inversion (Named as Exp_OCO3L, Exp_OCO2L and Exp_OCO3&2L, respectively). We compared the estimates of NEE and the evaluations against *in-situ* observations between the experiments with and without OG data, and found that assimilating OG data with our method can improve the inversions somewhat compared to removing OG.

We have added a paragraph in Section 4.5 in the revised manuscript (see Lines 439-462, Pages 20-21):

"In most of the previous studies that used OCO-2 XCO$_2$ to invert surface carbon fluxes, the OG data were not used (e.g., Peiro et al., 2022; Byrne et al., 2023), the reason is that the OG XCO$_2$ may have larger uncertainties, inversions assimilating OCO-2 OG retrievals produced unrealistic results of annual global ocean sinks (Peiro et al., 2022). In addition to its large uncertainties, we believe that another reason for the poor assimilation performance of OG is the relatively homogeneous distribution of XCO$_2$ on ocean, causing a large correlation of the model-data biases among different XCO$_2$ observations within a same region, which leads to observations at the same region having the same direction of adjustment for surface fluxes, and thus leads to a significant overestimated or underestimated of ocean carbon sink. Because of this, some assimilation algorithms (e.g., EnSRF) can only achieve better assimilation results when the model-data biases between observations have relatively small correlation or are uncorrelated. Therefore, in this study, we set the OG data with larger uncertainties than the LNLG data, and re-grided it at a coarser spatial resolution of 5°×5°. The results show that under this scheme, the inverted ocean sink is reasonable, with value of -2.6 PgC yr$^{-1}$ (Table 1). In addition, in order to compare the scheme that we have adopted in this study with the previous scheme that do not assimilate the OG, we added three additional inversion experiments, in which only the LNLG data were assimilated (Table S1). It could be found that all the three inversion experiments without OG observations place smaller constraints on the ocean fluxes compared to the original experiments, with the posterior ocean fluxes remaining almost identical to the prior ocean fluxes. Correspondingly, the inverted global land sink as well as the sinks in most regions show a slight decrease (Tables S2 and S3). Evaluations in comparison with in-situ observations showed that there are some increases in the a posteriori concentration biases for all three experiments after removing OG. For example, for the experiments assimilating OCO-2 data, the mean bias increased from 0.02 to 0.14 ppm (Table S4). This suggests that assimilating OG data with our method can improve the inversions somewhat compared to removing OG."

Line 134: Please explain the regridding process. Does the regridding process refer to the 'super-observation' described in section 2.1? How did the XCO$_2$ values and parameters for each sounding been processed? Did you take the mean, or median, or other methods? And can you justify the

method you used? How did you handle the outliers in the observations with one grid box? Also, for the 'super-observation', does it mean that for each model grid box, there's essentially only one observation being used by the model to constrain the posterior fluxes? If that's the case, why does the data amount (Figure 1) matter (except for the grids containing 0 OCO soundsing)?

**Response**: Thank you for this suggestion. The re-griding process was performed during the pre-processing of satellite data and does not involve the 'super-observation' process. The OCO observations are filtered using the parameter of $XCO_2$_quality_flag, which indicates the quality of the data. Only data with $XCO_2$_quality_flag equal 0 was selected. Then, the observations of LNLG were re-grided into $1° \times 1°$ grid cells, and those of OG were re-grided into $5° \times 5°$ using the arithmetic averaging method. The other variables like the column-averaging kernel and the retrieval error, which are provided along with the $XCO_2$ product, are also dealt with using the same method. This process is the same as Wang et al. (2019). For the 'super-observation', it mean that for each atmospheric transport model grid box, there's indeed only one observation being used by the model to constrain the posterior fluxes, but for each grid's flux, it is not only constrained by the observations of the grid it is on, because the atmosphere is moving and its downwind observations can all be used to constrain the flux of this grid, and in the system we use a two-layer localization scheme to select the surrounding and downwind observations that are used to constrain the flux of that grid. Therefore, the amount of observed data can have a significant impact on the inversion results. In the revised manuscript, we have further explained the 'super-observation' scheme (see Lines 148-151, Page 6) and also provided a detailed description of the localization technique (see Lines 152-166, Pages 6-7).

Line 209: Could you list out the annual $CO_2$ growth rates for 2020-2022 that you used to calculate the average growth rates?

**Response**: Thank you! When we conducted the inversion work, GCB2023 (i.e., Friedlingstein et al., 2023) had not yet been released, we used the 2020 and 2021 data (4.99 and 5.23 PgC/yr) from GCB2022, as well as the Annual Mean Global Carbon Dioxide Growth Rates (2.2 ppm) in 2022 reported by NOAA Global Monitoring Laboratory (https://gml.noaa.gov/ccgg/trends/gl_gr.html) by multi-by a factor of 2.124. The average atmospheric $CO_2$ growth rate is 4.96 PgC yr-1 for 2020-2022. We compared the results in GCB2022 and GCB2023 and found there are some differences in these values. In GCB2023, the $CO_2$ growth rates from 2020 to 2022 have been updated to 4.97016, 5.2038, and 4.63032 PgC/yr, with mean of 4.93 PgC/yr. Therefore, in the revised manuscript, we have updated this value to 4.93 PgC/yr.

Line 214: Why does the joint assimilation of OCO-2 and OCO-3 $XCO_2$ give the best performance on a global scale? One potential reason-spatial coverage of OCO-3 $XCO_2$ has been mentioned briefly in several places in the manuscript, but an in-depth discussion would be expected.

**Response**: Thank you for this suggestion. The OCO-3 satellite observations have a sufficient number of observations in the mid-latitude land region, while the OCO-2 satellite observations have a wide spatial coverage, even at high latitudes (Figure 1 in the original manuscript). Therefore, Exp_OCO3&2 assimilates sufficient observations in the mid-latitude region and observations in the high-latitude region, and has the advantages of OCO-2 and OCO-3 at the same time. At the same time, the joint assimilation of OCO-2 and OCO-3 $XCO_2$ also absorbs more observations than

assimilating the OCO-2 or OCO-3 alone, which will also make the assimilation better. Assimilating OCO-3 $XCO_2$ alone has poor performance, the reasons are that, on the one hand, the fact that it is only available between 52°S and 52°N, which leads to a lack of observational constraints on the carbon sinks at high latitudes, and there are the large fluctuations in the amount of observational data, which leads to significant differences in observational constraints at mid-latitudes at different times; on the other hand, its varied observation time also affect the inversions, but even choosing afternoon observations does not improve the inversions because the amount of observed data drops significantly. Therefore, a better option for the future would be to jointly assimilate the OCO-2 $XCO_2$ data and the OCO-3 $XCO_2$ retrievals observed in the afternoon (12:00 to 16:00 LST). We have added a detailed discussion about this issue in Section 4.5 of the revised manuscript (see Lines 463-506, Pages 22-23).

Line 221 - 224: Is the word 'sinks' in line 22 a typo? Otherwise the sentence does not make sense – the listed locations seem to have positive NEE values suggesting being $CO_2$ sources.
**Response**: Thank you! Yes, it is a typo. We have changed 'sinks' to 'sources' (see Line 283, Page 12).

Line 236: I would suggest the authors avoid using 'peaks' when describing the negative values to clear confusion, or maybe specify the values when doing comparison. For example, the 'peaks' for ExpOCO2 and Exp_OCO3&2 are higher than the prior when rotation 90 degrees for Figure 3 (f) and (i), but the actually corresponding values at the 'peaks' are lower because they are $CO_2$ sinks and the NEE values are negative. Same for 'the lowest peak' in line 237.
**Response**: Thank you! We have revised that sentence (see Lines 296-299, Page 13) as follows: "The posterior and prior fluxes have a similar distribution trend along the latitude, with a significant peak of carbon sink near 60°N, and the strongest sinks of Exp_OCO2 and Exp_OCO3&2 are comparable, which are significantly stronger than the a priori, while Exp_OCO3 has the weakest peak of carbon sink and that is close to the a priori."

Line 251: Potential confusion – by the word 'lower' do you mean the NEE value is lower (strong sinks) or the NEE value is higher (weaker sinks)?
**Response**: Many thanks for this suggestion. We mean that in all regions except temperate N. America, northern Africa, temperate Asia, and Australia, Exp_OCO3 shows a weaker carbon sink than Exp_OCO2. We have corrected it in the revised manuscript (see Line 309, Page 13).

Line 301: Which experiment are those numbers from?
**Response**: Thank you for this suggestion. These numbers are calculated by averaging all the 3 inversion experiments. We have revised that sentence to make it clear (See line 362, page 17).

Table 2 and Figure 4: Is the information presented in Table 2 and Figure 4 largely duplicated? If so, authors may consider removing Figure 4 if additional paragraphs are needed.
**Response**: Thank you for this suggestion. Figure 4 in the original manuscript is actually a visualization of the data in Table 2, so there is indeed a duplication of content. In the revised manuscript, we have removed Figure 4.

Figure 6, Figure 3 and Line 358: For high latitude areas (> 60 degree N), why is the BIAS from Exp_OCO3 not consistent with prior fluxes? Given the fact that no OCO-3 observations available beyond 52 degree north, I would expect the posterior fluxes are very similar to prior fluxes in high latitude areas since no observation can be used to constrain and optimize prior emissions, yet both Figure 3 and Figure 6 showed substantial changes when comparing posterior to prior from Exp_OCO3. It's possible that fluxes in high latitude can be updated due to spatial covariance assumed in the inversion system, therefore more details on the GCASv2 is needed in Section 2.1.

**Response**: Thank you for this suggestion. Since the atmosphere is moving, a change in flux at a certain location can cause a change in concentration downwind, i.e., observations downwind can sense the flux change at that location, and thus we can use observations downwind to constrain the flux in that area. In this study, we use a localization scale of 3000 km, which means that observations within a 3000km radius of a grid can be used to constrain the fluxes in that grid as long as they meet the localization requirements as described in section 2.1 in the revised manuscript.

We have added more information about the two-layer localization scheme (see Lines 152-166, Pages 6-7) as follows:

"There are inevitably spurious correlations in the EnKF method, to reduce the effect of spurious correlations, a two-layers localization scale was adopted in GCASv2, which is used to select which observations can be used for the flux analysis for each grid. The localization technique is based on the correlation coefficient between the simulated $XCO_2$ ensembles ($XCO_{2,i}^m$) in each observation location and the perturbed fluxes ($X_i^b$) in current model grids and their distances. The observations will be accepted for assimilation if the distance is less than 500 km and the correlation coefficient is greater than 0; if the distance is greater than or equal to 500 km and less than 3000 km and the correlation coefficient is significant ($p<0.05$), the observations will be accepted for assimilation. Otherwise, the observations are not accepted. The reason for this scheme is that considering the atmospheric horizontal diffusion, we believe that there must be a correlation between the flux of one grid and the concentrations in its neighbouring grids, and therefore observations are accepted as long as this correlation coefficient is greater than zero. In contrast, at distant locations (>500 km), where the effect of atmospheric horizontal diffusion is essentially negligible, the relationship between source and receptor is mainly due to atmospheric transport, and in order to minimize spurious correlations we require that such correlations must be significant."

Line 367: Could the bias exist prior? If there's no OCO-3 observation available in high latitudes, how can the OCO-3 observation introduce additional bias?

**Response**: Thank you for this suggestion. As the response in the previous comment, although the OCO-3 satellite has no observations at high latitudes, the observations downwind that area can be used to constrain the flux in that area. However, the assimilation of OCO-3 is much less effective compared to the OCO-2 satellite, which has observations in high latitudes, because only distant observations can be used in the Exp_OCO3 experiment.

Line 376: period '1'?

**Response**: Thank you for this suggestion. We mean the period from 1 August 2019 to 31 December 2022. We have corrected it in the revised manuscript (see Line 509, Page 23).